# Mutations in GRK2 cause Jeune syndrome by impairing Hedgehog and canonical Wnt signaling

Michaela Bosakova[1,2,3] (iD), Sara P Abraham[1], Alexandru Nita[1], Eva Hruba[3], Marcela Buchtova[3], S Paige Taylor[4], Ivan Duran[4], Jorge Martin[4], Katerina Svozilova[1,3], Tomas Barta[5], Miroslav Varecha[1], Lukas Balek[1], Jiri Kohoutek[6], Tomasz Radaszkiewicz[7] (iD), Ganesh V Pusapati[8,9], Vitezslav Bryja[7] (iD), Eric T Rush[10,11], Isabelle Thiffault[10,11], Deborah A Nickerson[12], Michael J Bamshad[12,13,14], University of Washington Center for Mendelian Genomics, Rajat Rohatgi[8,9], Daniel H Cohn[4,15], Deborah Krakow[4,16,17,*] (iD) & Pavel Krejci[1,2,3**] (iD)

## Abstract

**Mutations in genes affecting primary cilia cause ciliopathies, a diverse group of disorders often affecting skeletal development. This includes Jeune syndrome or asphyxiating thoracic dystrophy (ATD), an autosomal recessive skeletal disorder. Unraveling the responsible molecular pathology helps illuminate mechanisms responsible for functional primary cilia. We identified two families with ATD caused by loss-of-function mutations in the gene encoding adrenergic receptor kinase 1 (ADRBK1 or GRK2). GRK2 cells from an affected individual homozygous for the p.R158* mutation resulted in loss of GRK2, and disrupted chondrocyte growth and differentiation in the cartilage growth plate. GRK2 null cells displayed normal cilia morphology, yet loss of GRK2 compromised cilia-based signaling of Hedgehog (Hh) pathway. Canonical Wnt signaling was also impaired, manifested as a failure to respond to Wnt ligand due to impaired phosphorylation of the Wnt co-receptor LRP6. We have identified GRK2 as an essential regulator of skeletogenesis and demonstrate how both Hh and Wnt signaling mechanistically contribute to skeletal ciliopathies.**

**Keywords** asphyxiating thoracic dystrophy; GRK2; hedgehog; smoothened; Wnt

**Subject Categories** Development; Genetics, Gene Therapy & Genetic Disease; Musculoskeletal System

## Introduction

A single primary cilium protrudes from nearly every post-mitotic vertebrate cell, and cilia sense and transduce a vast array of extracellular cues. Cilia utilize intraflagellar transport (IFT), a bidirectional system that builds and maintains the cilium while also facilitating protein entry, exit and trafficking through the organelle. IFT is governed by a large multimeric protein complex with two main subcomplexes, IFT-A and IFT-B. The anterograde IFT is driven by the kinesin motor KIF3 and mediates transport from the base to the tip of cilia, while retrograde IFT is driven by the dynein-2 motor and transports cargo from the tip to the base of the cilium (Kozminski et al, 1993). Vertebrate primary cilia act as signaling centers for the Hedgehog (Hh) family of morphogens

---

1 Department of Biology, Faculty of Medicine, Masaryk University, Brno, Czech Republic
2 International Clinical Research Center, St. Anne's University Hospital, Brno, Czech Republic
3 Institute of Animal Physiology and Genetics of the CAS, Brno, Czech Republic
4 Department of Orthopaedic Surgery, David Geffen School of Medicine at UCLA, Los Angeles, CA, USA
5 Department of Histology and Embryology, Faculty of Medicine, Masaryk University, Brno, Czech Republic
6 Veterinary Research Institute, Brno, Czech Republic
7 Institute of Experimental Biology, Faculty of Science, Masaryk University, Brno, Czech Republic
8 Department of Biochemistry, Stanford University, Palo Alto, CA, USA
9 Department of Medicine, Stanford University, Palo Alto, CA, USA
10 Children's Mercy Kansas City, Center for Pediatric Genomic Medicine, Kansas City, MO, USA
11 Department of Pediatrics, University of Missouri, Kansas City, MO, USA
12 Department of Genome Sciences, University of Washington, Seattle, WA, USA
13 Department of Pediatrics, University of Washington, Seattle, WA, USA
14 Division of Genetic Medicine, Seattle Children's Hospital, Seattle, WA, USA
15 Department of Molecular Cell and Developmental Biology, University of California at Los Angeles, Los Angeles, CA, USA
16 Department of Human Genetics, David Geffen School of Medicine at UCLA, Los Angeles, CA, USA
17 Department of Obstetrics and Gynecology, David Geffen School of Medicine at UCLA, Los Angeles, CA, USA
    *Corresponding author. Tel: +13109831252; E-mail: dkrakow@mednet.ucla.edu
    **Corresponding author. Tel: +420549495395; E-mail: krejcip@med.muni.cz

and also orchestrate a variety of other cell signaling cascades (Gerhardt *et al*, 2016).

Mutations in genes affecting primary cilia structure or function cause ciliopathies, a group of pleiotropic disorders. Among these disorders is a subset with profound abnormalities in the skeleton, termed skeletal ciliopathies. Asphyxiating thoracic dystrophy (ATD) or Jeune syndrome is considered an autosomal recessively inherited skeletal ciliopathy characterized by a long narrow chest, shortened long bones, and occasional polydactyly. Other affected organs can include the brain, retina, lungs, liver, pancreas, and kidneys. Characteristic radiographic findings include handlebar shaped clavicles, short horizontal ribs, shortened appendicular bones with irregular metaphyseal ends, a small pelvis with a trident shaped acetabular roof, and brachydactyly with cone-shaped epiphyses. Other skeletal ciliopathies with overlapping features include short rib polydactyly syndrome (SRPS), Ellis-van Creveld dysplasia (EVC), and cranioectodermal dysplasia (CED). ATD is clinically and radiographically most similar to the perinatal-lethal SRPS. However, while ATD is not uniformly lethal, long-term survivors often have multi-organ system complications. Mutations in genes affecting ciliogenesis or the IFT process produce a broad phenotypic spectrum of skeletal ciliopathies, and there is significant allelic and locus heterogeneity. Many of the genes mutated in this spectrum of disorders encode IFT-A proteins and their motors: the cytoplasmic dynein-2 motor heavy chain, DYNC2H1 (MIM 603297), WDR34 (MIM 615633), WDR60 (MIM 615462), WDR19 (also known as IFT144) (MIM 614376), IFT140 (MIM 266920), WDR35 (also known as IFT121) (MIM 614091), TTC21B (also known as IFT139) (MIM 612014), IFT43 (MIM 614068), TCTEX1D2 (MIM 617353), IFT122 (NIM 606045), IFT140 (MIM 614620), and DYNC2LI1 (MIM 617083) (Gilissen *et al*, 2010; Walczak-Sztulpa *et al*, 2010; Arts *et al*, 2011; Bredrup *et al*, 2011; Davis *et al*, 2011; Perrault *et al*, 2012; Huber *et al*, 2013; McInerney-Leo *et al*, 2013; Schmidts *et al*, 2013, 2015; Taylor *et al*, 2015). Mutations also have been reported in the genes encoding IFT-B members: IFT80 (MIM 611263), IFT52 (MIM 617094), IFT81 (MIM 605489), and IFT172 (MIM 615630) (Beales *et al*, 2007; Halbritter *et al*, 2013; Duran *et al*, 2016; Zhang *et al*, 2016). Additional locus heterogeneity in these disorders results from mutations in the genes that encode the centrosomal kinase NEK1 (MIM 604588), MAP kinase family member ICK (MIM 612325), CEP290 (MIM 6101142), KIAA05866 (MIM 610178), and C21ORF2 (Thiel *et al*, 2011; Alby *et al*, 2015; Paige Taylor *et al*, 2016; McInerney-Leo *et al*, 2017), as well as two proteins involved in planar cell polarity, INTU (MIM 610621) and FUZ (MIM 610622) (Toriyama *et al*, 2016; Zhang *et al*, 2018); three centrosomal proteins, EVC (NIM 604831), EVC2 (NIM 607261) and CEP120 (NIM 613446) (Ruiz-Perez *et al*, 2000; Galdzicka *et al*, 2002; Shaheen *et al*, 2015); and the nuclear membrane protein LBR (NIM 600024) (Zhang *et al*, 2018).

In an effort to identify additional ATD genes and increase our understanding of ciliary function in skeletal development, exome sequence analysis was carried out in a cohort of patients within the skeletal ciliopathy spectrum, and two families were identified with pathogenic variants in the adrenergic receptor kinase beta 1 gene (*ADRBK1* or *GRK2*). GRK2 is one of seven G protein-coupled receptor kinases (GRKs), which bind to and rapidly desensitize activated G protein-coupled receptors (GPCRs). GPCRs localize to the cell membrane where they sense extracellular cues and, through coupling with G proteins, relay signals to the cell. The wide repertoire of GPCR ligands includes photons, peptides, hormones, lipids, and sugars. With over 800 known members, the GPCR family represents the largest and most diverse class of eukaryotic membrane receptors (Fredriksson *et al*, 2003). In its canonical signaling cascade, GRK2 is activated by protein kinase A and phosphorylates the beta-adrenergic receptor, desensitizing it, and thus preventing the cells from overstimulation by catecholamines (Hausdorff *et al*, 1990). In addition to GPCR signaling, the ubiquitously expressed GRK2 participates in many other processes, including regulation of phospholipase C, PI3 kinase, and RAF kinase activity, modulation of cytoskeletal proteins and activation of Smad and NFκB signaling through direct phosphorylation (Evron *et al*, 2012).

GRK2 also functions in the Hh pathway. Hh signaling is initiated by Hh ligand binding to the Patched 1 (PTCH1) receptor, followed by phosphorylation-dependent accumulation of Smoothened (SMO) GPCR at the cell membrane (*Drosophila*) or in the primary cilia (vertebrates). This results in activation of the Hh gene expression through degradation of the GLI2/GLI3 transcriptional repressors and production of GLI activators (Forbes *et al*, 1993; Van den Heuvel & Ingham, 1996; Dai *et al*, 1999; Sasaki *et al*, 1999; Aza-Blanc *et al*, 2000; Wang & Holmgren, 2000; Haycraft *et al*, 2005; Rohatgi *et al*, 2007; Tukachinsky *et al*, 2010; Su *et al*, 2011). The ciliary cAMP levels appear to have a central role in this process, regulating activity of protein kinase A (PKA) that is critical for formation of GLI repressors (Humke *et al*, 2010; Tuson *et al*, 2011; Mukhopadhyay *et al*, 2013; Niewiadomski *et al*, 2014). In this context, GRK2 acts as a positive regulator of Hh signaling, necessary for maximal Hh response in both *Drosophila* and vertebrates (Jia *et al*, 2004; Maier *et al*, 2014; Li *et al*, 2016); yet, the molecular mechanisms are not fully understood. GRK2 was shown to phosphorylate SMO C-tail, leading to ciliary accumulation of SMO and Hh pathway activity (Chen *et al*, 2011); the former has been disputed and appears cell context dependent (Zhao *et al*, 2016; Pusapati *et al*, 2018). GRK2-mediated SMO phosphorylation was shown to induce β-arrestin (ARRB) recruitment, leading to KIF3A-dependent cilia accumulation of SMO (Chen *et al*, 2004; Kovacs *et al*, 2008); however, later studies showed normal cilia accumulation of SMO in ARRB knock-out cells (Pal *et al*, 2016; Desai *et al*, 2020). Ligand-dependent removal of the Hh pathway inhibitor GPR161 was shown to depend on GRK2 activity (Pal *et al*, 2016; Pusapati *et al*, 2018). More recently, the GRK2 activity was shown necessary for SMO to sequester the catalytic subunit of PKA which in turn can no longer phosphorylate GLI3 in order to produce the GLI3 repressor (Niewiadomski *et al*, 2014; preprint: Happ *et al*, 2020).

Despite the limited understanding of the complex role of GRK2 in Hh signaling, its depletion by RNA interference in cell cultures clearly inhibits Hh activity (Meloni *et al*, 2006; Chen *et al*, 2010, 2011; Maier *et al*, 2014), and the *Grk2*$^{-/-}$ NIH3T3 do not respond to Hh stimulation as they fail to degrade GLI3 repressor and to activate Hh gene expression (Zhao *et al*, 2016; Pusapati *et al*, 2018). This effect is recapitulated by morpholino-mediated knock-down of *grk2* and in the maternal-zygotic *grk2* mutant zebrafish embryos (Philipp *et al*, 2008; Evron *et al*, 2011; Zhao *et al*, 2016). Loss of *grk2* in zebrafish results in a curved body axis, U-shaped body somites and severe cyclopia (Zhao *et al*, 2016), phenocopying the *smo* mutant (Chen *et al*, 2001). In contrast, the Grk2 mouse knock-out shows milder phenotypes, at least in the neural tube patterning, yet is lethal around midgestation (Jaber *et al*, 1996; Philipp *et al*, 2008). This lethality was associated

with developmental heart defects (Jaber *et al*, 1996; Matkovich *et al*, 2006), yet the timing did not allow for studying the effect of Grk2 loss on later developing tissues and organs such as skeleton. Herein, we demonstrate that pathogenic variants in *GRK2* produce ATD and modulate both Hh and Wnt signaling, demonstrating that GRK2 is an essential regulator of skeletogenesis.

## Results

### Loss of GRK2 results in ATD

The first proband (R05-365A) was born at 38 weeks to second-cousin parents. Prenatal ultrasound showed shortened limbs with a lag of approximately 8–9 weeks from the estimated due date. The pregnancy was complicated by ascites and hydrops fetalis that arose in the third trimester. The proband was delivered at term and had a very small chest with underlying pulmonary insufficiency. Additionally, she had low muscle tone, an atrial septal defect, hypoplastic

nails, but no polydactyly. Radiographic findings included long narrow clavicles, short horizontal bent ribs with lack of normal distal flare, short humeri, mesomelia with bending of the radii, short femora and tibiae with broad metaphyses, diminished mineralization, and no endochondral ossification delay (Fig 1A and C). She expired 5 days after birth. The findings compared to characteristic ATD are delineated in Table 1.

In the second family, the elder of the two affected female siblings (proband Cmh001543-01) was delivered at 39 weeks' gestational age by cesarean section. The pregnancy was complicated by exposure to cannabis and the antiseizure medication, levetiracetam. Prenatal ultrasound at 20 weeks showed shortened long bones and a small chest and evaluation of the chest to abdominal circumference was predictive of lethality. At delivery, the proband was cyanotic with Apgar scores were $3^1$, $7^5$, and $8^{10}$ and the head circumference was 32.5 cm, birth length 46 cm, and weight 2.82 kg. She was intubated at 12 min of life and initially required high levels of respiratory support and was extubated on day 14 of life, but continued to have chronic respiratory issues. Radiographic findings showed similar

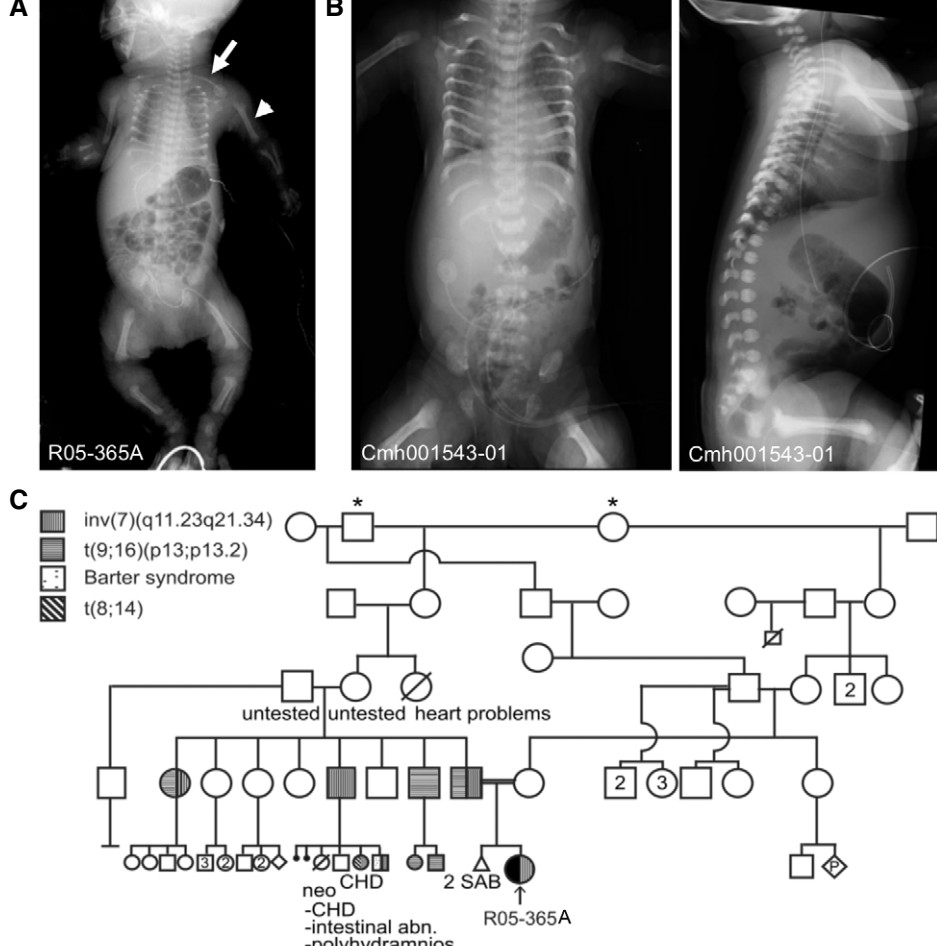

**Figure 1.  Asphyxiating thoracic dystrophy (ATD) probands R05-365A and Cmh001543-01.**

A   AP radiograph demonstrates characteristic findings of ATD in the R05-365A proband. Note the shortened humeri (closed arrowhead) and elongated clavicles (arrow).

B   Radiographs of the Cmh001543-01 proband showing similar findings.

C   Family R05-365A pedigree; * indicates common ancestors. CHD, congenital heart disease, SAB, spontaneous abortion. Abn, abnormalities.

Table 1. Clinical and radiographic phenotype of ATD and the R05-365A and Cmh001543-01 and -02 cases.

| Clinical and Radiographic Phenotype | ATD/Jeune syndrome | R05-365A | Cmh001543-01 and -02 |
|---|---|---|---|
| Autosomal recessive | + | + | + |
| Retinal insufficiency (age dependent) | + | − | − |
| Pulmonary insufficiency/hypoplasia | + | + | + |
| Polycystic liver disease/hepatic fibrosis (age dependent) | + | − | − |
| Cystic kidneys/chronic renal failure (age dependent) | + | − | − |
| Congenital heart defect | + | + | − |
| Ascites | + | + | − |
| Lethality due to pulmonary hypoplasia | + | + | + |
| Long narrow thorax | + | + | + |
| Short horizontal ribs | + | + | + |
| Handlebar clavicles | + | − | + |
| Small pelvis | + | + | + |
| Hypoplastic iliac wings (infancy) | + | + | + |
| Short long bones | + | + | + |
| Irregular metaphyses | + | + | + |
| Short fibulae (relative) | + | + | Unknown |
| Short ulnae (relative) | + | + (bowed) | Unknown |
| Brachydactyly | + | Unknown | Unknown |
| Cone-shaped epiphyses (childhood) | + | Unknown | Unknown |
| Occasional polydactyly | + | − | − |

R05-365A, International Skeletal Dysplasia Registry Number, +, present, −, absent.

findings to the first proband (R05-365A) and included handlebar clavicles, short horizontal bent ribs with lack of normal distal flare, short humeri, broad irregular metaphyses, diminished mineralization, axial clefts, and odontoid hypoplasia (Fig 1B). She died at 5.5 months of age related to respiratory sequelae due to chronic lung disease. The second affected sibling had similar radiographic findings, but did not require prolonged respiratory support. She had questionable seizure events, but otherwise exhibited a much less severe clinical course and was alive at 12 months of age.

Using exome sequence analyses of the R05-365A proband, homozygosity for a nonsense variant in exon 6 of *GRK2*, c. 469 C>T predicting the amino acid change p.R158*, was identified. The pathogenic variant localizes to the G protein signaling (RGS) domain of GRK2 (Fig 2A and C). The pathogenic variant occurred within a 13 Mb block of homozygosity on chromosome 11 and has not been seen in population databases. Detection of GRK2 expression, by RT–PCR of cDNA and Western blot analysis of protein, respectively, demonstrated loss of both GRK2 transcript and protein in cultured patient fibroblasts (Fig 2D and E). The data thus demonstrate that the p.R158* pathogenic variant results in a *GRK2* null ($GRK2^{-/-}$) genotype.

In this second family, exome analysis identified two pathogenic variants, each inherited from an unaffected parent. The maternally inherited variant, c.1348_1349del in exon 16, predicted an amino acid substitution followed by a frameshift, p.Ser450Profs*40. The altered serine residue localizes to the kinase domain. The paternally inherited variant, c.555 +1 G>A, is predicted to lead to exon 7 skipping and introduction of a frameshift in exon 8, p.L186*. This change resides in a region between the RGS and kinase domain of GRK2 (Fig 2B and C). Both of these variants were absent in control databases. These biallelic alterations are predicted to lead to nonsense-mediated decay of the transcript and loss of the GRK2 protein. Cell lines were not available from either affected individual in this family.

## GRK2 is essential for skeletal growth plate development

Although the early embryonic lethality prevents detailed analyses of the skeleton, the $Grk2^{-/-}$ mice embryos display growth retardation, suggesting GRK2 role in skeletal development (Jaber *et al*, 1996; Philipp *et al*, 2008). The profound skeletal abnormalities observed as a result of loss of GRK2 in the R05-365A patient suggested that the function of the protein is essential for normal development of the cartilage growth plate. To define the localization of GRK2 during embryonic skeletal development, immunohistochemistry was performed on human cartilage growth plate derived from a control 17-week gestational age distal femur. GRK2 was present throughout the growth plate and adjacent tissues, with the strongest staining detected in proliferating chondrocytes, primary spongiosum, and perichondrium (Fig 3A).

To investigate the effects of GRK2 depletion on skeletal development through the cartilage growth plate, we compared the microanatomical appearance of femoral growth plate cartilage between the affected (R05-365A) individual and a 17-week fetal control. Compared with control, the affected growth plate architecture was profoundly disturbed, with a markedly expanded reserve zone containing small round chondrocytes, absence of characteristic proliferating chondrocytes, an extremely short hypertrophic zone,

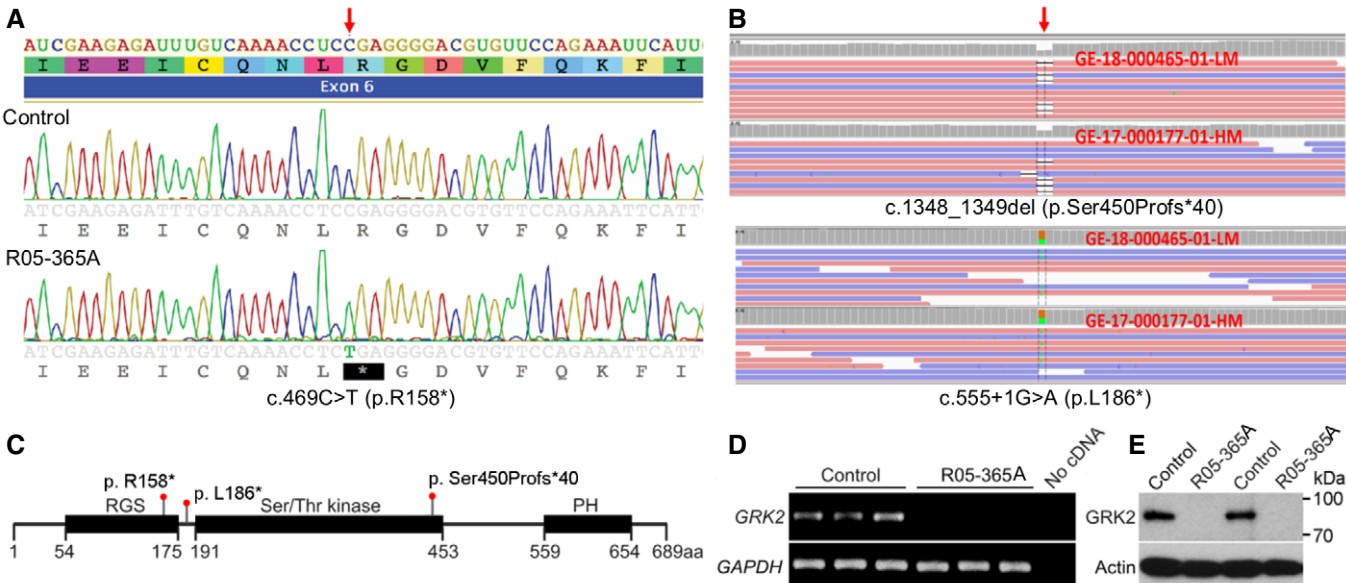

**Figure 2. Homozygosity and compound heterozygosity for null mutations in *GRK2* in ATD.**

A   Sanger sequence trace showing homozygosity for the c.469C>T mutation, predicting a stop codon in exon 6 of *GRK2* (arrow).
B   Exome analysis showing the biallelic changes c. 1348_1349del and c.555 +1G>A. Red arrow points to the place of mutation. Orange and green boxes indicate the reference nucleotide and its substitution, respectively.
C   Domain composition of GRK2 and the corresponding location of the mutations (RGS, regulator of G protein signaling domain; PH, pleckstrin homology domain).
D   Expression of *GRK2* as measured by RT–PCR in three replicates each of control and R05-365A fibroblasts, demonstrating absence of *GRK2* transcript in ATD cells. *GAPDH* served as a positive control.
E   Two control and ATD fibroblast samples were immunoblotted for GRK2. Note the absence of GRK2 protein in the ATD cells. Actin served as a loading control. The data shown are representative of three independent experiments.

Source data are available online for this figure.

small hypertrophic chondrocytes, and a minimal zone of terminal differentiation (Fig 3B). The profound disorganization of the growth plate cartilage underlies the skeletal changes causing ATD and demonstrates that, at the cellular level, the loss of GRK2 affects chondrocyte proliferation and hypertrophic differentiation. To further confirm the role of GRK2 in regulation of bone development, we inhibited GRK2 kinase activity by CMPD101, a chemical inhibitor of GRK2 activity [32]. CMPD101 was injected into the right wing bud of chick embryos at stage HH20-22, and the effect on skeletal development was determined 10–12 days later. CMPD101 caused no gross abnormalities in the shape of ulna, radius and humerus, but lead to shortening of all three skeletal elements (Fig 3C).

The femoral growth plate cartilage samples available on *GRK2*⁻/⁻ individual (R05-365A) were stained with alcian blue, to visualize the proteoglycan content of the chondrocyte extracellular matrix. At pH2.5 (proteoglycan staining) (Green & Pastewka, 1974), no significant change was found in the signal between control and R05-365A growth plate. At pH1.0 (sulfated proteoglycan staining), a lack of signal was found in R05-365A (Fig 3D). The putative lack of proteoglycan sulfation in the absence of GRK2 was addressed in two independent *in vitro* models. First, the mesenchymal cultures were established from the wing buds of stage HH20 chicken embryos, and differentiated into the cartilage in a 5-day micromass culture (Horakova *et al*, 2014). Treatment with CMPD101 led to a significantly reduced alcian blue staining at both pH2.5 and pH1.0 (Fig 3E;

graphs, cartilage nodules staining). Second, the *Grk2*⁻/⁻ NIH3T3 fibroblasts (Pusapati *et al*, 2018) were differentiated to cartilage in a 7-day micromass culture and stained with alcian blue to estimate the sulfated proteoglycan content. A significant reduction of alcian blue staining was found at both pH2.5 and pH1.0, with more profound at pH1.0 (Fig 3F). Our data suggest a role of GRK2 in regulation of sulfation of the cartilage extracellular matrix proteoglycans.

**GRK2 is not required for ciliogenesis**

Because ATD is a ciliopathy, we next examined the affected patient's *GRK2*⁻/⁻ patient fibroblasts for defects in ciliogenesis. Cells were serum starved to produce primary cilia, and immunohistochemistry was performed to visualize the axoneme (acetylated tubulin), ciliary membrane (ARL13B), and basal bodies (pericentrin) (Fig EV1A). No differences in ARL13B or acetylated tubulin staining were found between R05-365A and control fibroblasts. Because tubulin-based staining does not visualize ciliary tips, ARL13B signal was used to measure cilia length. We found no significant differences in the average cilia length between control and *GRK2*⁻/⁻ cells (Fig EV1B). Similarly, no differences were seen in the efficiency of ciliogenesis, determined by counting the number of ciliated cells after starvation (Fig EV1C).

To determine whether loss of GRK2 affected IFT in the normally appearing primary cilia, we compared the distribution of

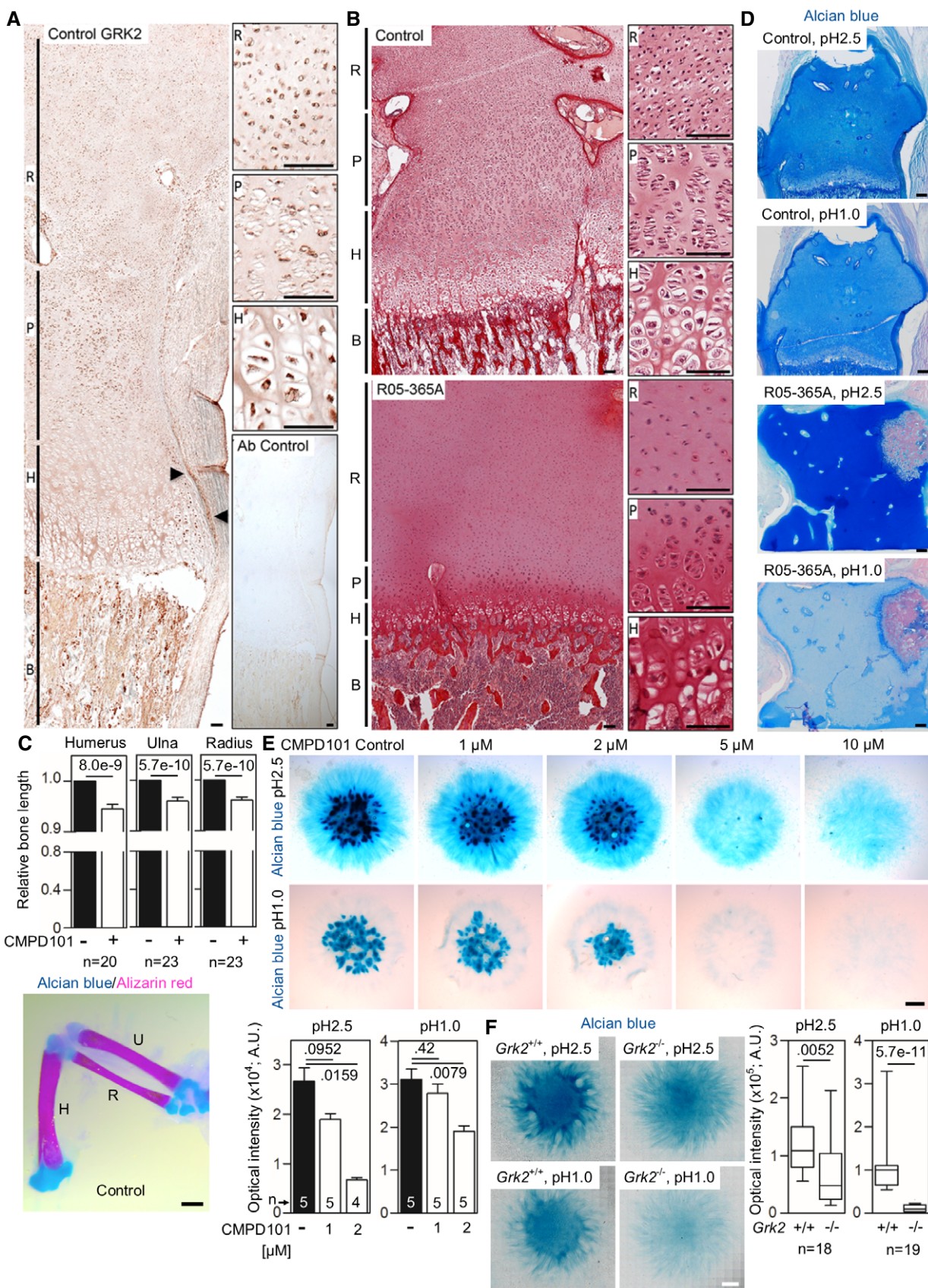

**Figure 3.**

◀

**Figure 3. GRK2 is essential for skeletal growth plate development.**

A GRK2 immunohistochemistry in a control 17-week gestational age distal femur growth plate demonstrating widespread expression, particularly in the bone (B), proliferating zone of the growth plate (P), and the perichondrium (arrowheads). Non-reactive antibody was used as a control (Ab control). R, reserve zone. H, hypertrophic zone. Scale bars, 100 μm.

B Picrosirius red staining of the femoral growth plates of a control and the ATD R05-365A fetus. The control growth plate shows normal architecture with clearly distinguishable reserve (R), proliferating (P) and hypertrophic (H) zones. A profound disruption of growth plate architecture is obvious in the $GRK2^{-/-}$ growth plate, which shows irregularly shaped reserve zone chondrocytes, a lack of discernable proliferating chondrocytes, and a significantly under-developed hypertrophic zone. Scale bars, 100 μm.

C Shortened long bones in the chicken wings injected with CMPD101. The right limb bud was injected with CMPD101, and the left one was left as a control. Note the shortened humerus (H), ulna (U), and radius (R) in the CMPD101-injected wing. Mean ± SEM. Mann–Whitney *U*-test; number of animals is indicated. Scale bar, 150 μm.

D Alcian blue staining of the femoral growth plates of a control and the ATD R05-365A fetus. Note the significantly lesser blue staining of the R05-365A growth plate stained by alcian blue pH 1.0, suggesting impaired sulfation of the extracellular matrix. Scale bars, 200 μm.

E Alcian blue staining of the chicken limb bud micromasses treated with CMPD101. Note the gradual inhibition of the cartilage nodules with increasing concentration of CMPD101, quantified below. Mean ± SEM. Mann–Whitney *U*-test; number of micromasses is indicated. A representative experiment of four (pH 2.5) and three (pH 1.0) is shown. Scale bar, 100 μm.

F Alcian blue staining of the micromasses generated from $Grk2^{+/+}$ and $Grk2^{-/-}$ NIH3T3 cells. Note the mildly weaker staining by alcian blue pH 2.5 and the nearly absent sulfation stained by alcian blue pH 1.0 in the $Grk2^{-/-}$ micromasses. Central band, median. Box, 1$^{st}$-3$^{rd}$ quartile. Whiskers, 10%-90% percentile. Mann–Whitney *U*-test; number of micromasses is indicated. Scale bar, 1 mm.

Source data are available online for this figure.

several IFT components between control and $GRK2^{-/-}$ cilia (Fig EV1D). The immunostaining for IFT43 (IFT-A complex component), WDR34 (IFT-A dynein-2 motor complex member), IFT88 (IFT-B complex component), TRAF3IP1 (IFT-B complex component), and KIF3A (IFT-B kinesin motor component) showed that the IFT components accessed and localized to cilia similarly between control and $GRK2^{-/-}$ cells. The distribution of ICK (centrosomal kinase) and GLI3 (Hh target transcription factor) in $GRK2^{-/-}$ cilia also followed the reported normal localization to the basal bodies or the tips of the cilia, respectively (Paige Taylor *et al*, 2016; Fig EV1A and D).

Finally, we used two chondrocyte cell lines established from control fetal growth plate cartilage (lines R00-082 and R92-284), and treated them with chemical inhibitors of GRK2 kinase activity, paroxetine, and CMPD101 (Thal *et al*, 2012; Pusapati *et al*, 2018), and examined for number of ciliated cells and for cilia length; no effect on cilia structure, length, or frequency was found (Fig EV1E–G). Together, these data demonstrated that loss of GRK2, or reduction of GRK2 kinase activity, produces no effect on ciliogenesis, ciliary length, or cilia morphology.

### $GRK2^{-/-}$ cells do not respond to Hh pathway activation and fail to traffic Smoothened to cilia

The data shown in Fig EV1 demonstrated that loss of GRK2 did not affect ciliogenesis or localization of IFT components, and thus, we hypothesized that the defect causing ATD is likely to lie in the specialized signaling functions of primary cilia. In vertebrates, the core components of the Hh pathway localize to the primary cilia. In the absence of stimulation, the cilia-localized Hh receptor PTCH1 prevents SMO from accumulating in cilia (Ingham *et al*, 1991; Rohatgi *et al*, 2007). Under this condition, the transcriptional effector of the Hh pathway, GLI3, is proteolytically cleaved into the repressor form, GLI3$^{R}$. Upon Hh binding to PTCH1, SMO accumulates in primary cilia, resulting in priming of the ciliary GLI3 for degradation. This relieves the GLI3$^{R}$-mediated repression on Hh target genes, leading to their transcriptional induction (Dai *et al*, 1999; Aza-Blanc *et al*, 2000; Wang & Holmgren, 2000).

To evaluate the cellular response to Hh pathway activation, control and patient $GRK2^{-/-}$ fibroblasts were treated with Smoothened agonist (SAG) (Chen *et al*, 2002), and whole cell extracts were immunoblotted for GLI3. In control cells, SAG treatment resulted in the progressive downregulation of both full-length GLI3 (GLI3$^{FL}$) and GLI3$^{R}$, demonstrating normal processing (Wen *et al*, 2010). There was no effect of SAG in $GRK2^{-/-}$ cells, which maintained constant GLI3$^{FL}$ and GLI3$^{R}$ levels throughout the 24 h of SAG treatment (Fig 4A). Because GLI3$^{FL}$ acts as a transcriptional activator, and skeletal ciliopathies often show altered GLI3$^{FL}$ to GLI3$^{R}$ ratios (Murdoch & Copp, 2010), we quantified the amounts of GLI3$^{FL}$ and GLI3$^{R}$ in cells. When treated with SAG, control fibroblasts showed an increased ratio of GLI3$^{FL}$ to GLI3$^{R}$ compared to baseline; this ratio was not increased in $GRK2^{-/-}$ cells (mean ± SEM, 2.23 ± 0.31 in control vs. 0.94 ± 0.35 in R05-365A; Mann–Whitney *U*-test) (Fig 4B).

To test whether reduced GLI3 processing in SAG-treated $GRK2^{-/-}$ cells resulted in defective induction of Hh target genes, cells were treated with SAG for 24 h and transcript levels of the GLI3 target genes *GLI1* and *PTCH1* (Dai *et al*, 1999; Ågren *et al*, 2004) were determined by quantitative RT–PCR. Fig 4C demonstrates a potent, SAG-mediated induction of *GLI1* and *PTCH1* expression in control cells, in contrast to $GRK2^{-/-}$ cells, which did not induce *GLI1* or *PTCH1* expression in response to SAG. These data demonstrate that loss of GRK2 compromises Hh signaling in cells, rendering them unresponsive to Hh pathway activation.

While SMO represents an atypical GPCR, it is phosphorylated by GRKs in a similar way as classic GPCRs. In response to Hh ligands, GRK2 phosphorylates SMO at multiple C-terminal sites, and this phosphorylation induces conformational changes in SMO which facilitate interactions with ARRB and the kinesin KIF3A that were reported necessary for SMO translocation to cilia (Chen *et al*, 2004; Meloni *et al*, 2006; Kovacs *et al*, 2008). The SMO phosphorylation was inhibited in R05-365A fibroblasts, as shown by phosphoshift assay (Jia *et al*, 2004; Zhang *et al*, 2004; Chen *et al*, 2011) in cells transfected by murine SMO (Fig 4D). Removal of GRK2-induced SMO phosphorylation sites has been shown to abolish SMO translocation to primary cilia in response to Hh ligands (Chen *et al*, 2011).

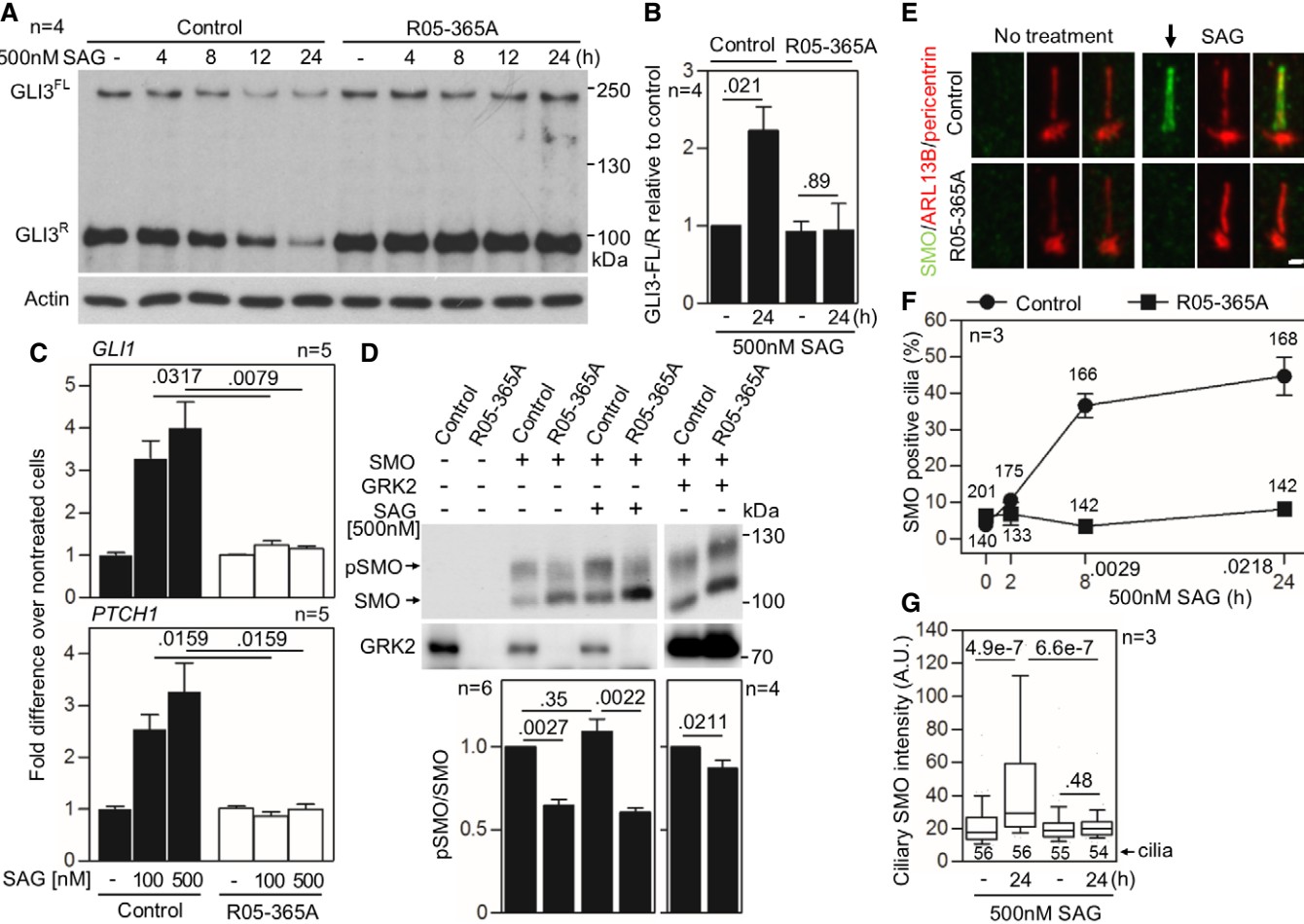

**Figure 4. GRK2$^{-/-}$ cells do not respond to Hh pathway activation and fail to accumulate Smoothened in cilia.**

A, B  Defective GLI3 processing in GRK2$^{-/-}$ fibroblasts. Cells were serum starved to induce cilia before treatment with the Hh pathway activator SAG (Smoothened agonist), and the whole cell lysates were immunoblotted for GLI3. Note the SAG-mediated downregulation of both full-length (FL) and repressor (R) GLI3 in control fibroblasts, demonstrating normal processing. There was no effect of SAG on GLI3 processing in GRK2$^{-/-}$ cells. A representative experiment of four is shown. (B) Quantitation of the GLI3$^{FL}$ to GLI3$^{R}$ ratio using densitometry. Mean ± SEM. Mann–Whitney $U$ test; number of biological replicates is indicated.

C  Defective SAG response in GRK2$^{-/-}$ fibroblasts. Cells were treated with SAG for 24 h. Transcript levels of GLI3 targets GLI1 and PTCH1 were determined by qRT–PCR; GAPDH was used for normalization. Note the SAG-mediated induction of GLI1 and PTCH1 expression in controls but not in GRK2$^{-/-}$ cells. Mean ± SEM. Mann–Whitney $U$-test; number of biological replicates is indicated.

D  Under-phosphorylation of Smoothened (SMO) in GRK2$^{-/-}$ fibroblasts. Lysates of SMO-transfected cells were resolved by phospho(p)-shift PAGE and immunoblotted for SMO. The portion of pSMO analyzed by densitometry is shown. GRK2 add-back demonstrates a rescue of the pSMO in GRK2$^{-/-}$ fibroblasts. Mean ± SEM. Mann–Whitney $U$-test; number of biological replicates is indicated.

E–G  Defective SMO accumulation in cilia of GRK2$^{-/-}$ fibroblasts. Cells were treated with SAG and immunostained for SMO, ARL13B, and pericentrin. Note the failure in SMO accumulation in cilia of GRK2$^{-/-}$ cells (arrow). Scale bar, 2 μm (E). The percentage of SMO-positive cilia was calculated and plotted. Mean ± SEM. Welch's $t$-test; number of biological experiments and the total numbers of analyzed cilia are indicated (F). The intensity of ciliary SMO was analyzed and plotted. Central band, median. Box, 1$^{st}$-3$^{rd}$ quartile. Whiskers, 10%-90% percentile. Mann–Whitney $U$-test; number of biological experiments and the total numbers of analyzed cilia are indicated (G).

Source data are available online for this figure.

To determine whether this mechanism is relevant in GRK2$^{-/-}$ cells, we treated the cells with SAG, and the immunocytochemistry was employed using SMO, ARL13B, and pericentrin to visualize SMO translocation into cilia (Fig 4E). In control human fibroblasts, SAG induced progressive accumulation of SMO in cilia, which was observed in nearly 50% of cells at 24 h of SAG treatment. In contrast, no SMO accumulation in primary cilia was observed in GRK2$^{-/-}$ cells (Fig 4F). This was confirmed by measurement of

SMO signal intensity in the cilia; significant accumulation of SMO signal was found in SAG-treated control cells but not in GRK2$^{-/-}$ patient cells (Fig 4G).

To further confirm the data obtained with patient GRK2$^{-/-}$ fibroblasts, we used two chondrocyte cell lines (lines R00-082 and R92-284). Inhibition of GRK2 kinase activity in these cells, via treatment with CMPD101 and paroxetine, did not affect ciliogenesis or ciliary length (Fig EV1E–G). To assess SAG response in

CMPD101- and paroxetine-treated chondrocytes, the cells were starved for 24 h in the presence of these inhibitors before treatment with SAG for additional 12 h. Cilia were stained by ARL13B and SMO antibodies, and the intensity of SMO signal in cilia was determined. Both GRK2 inhibitors significantly impaired SAG-mediated SMO accumulation in cilia (Figs 5A and B, and EV2A and B), similar to the findings in the $GRK2^{-/-}$ patient cells (Fig 4E–G). Also similar to patient cells, the SAG-mediated increased GLI3$^{FL}$ to GLI3$^{R}$ ratio and upregulated GLI1 expression was inhibited in cells treated with CMPD101 or paroxetine (Figs 5C and D, and EV2C and D). Next, the doxycycline (DOX)-inducible shRNA expression was used to target GRK2 in control human R00-082 fibroblasts. Cells were transduced with lentiviral vectors for stable integration, and probed for GRK2 expression after 4 days of DOX induction. Approximately 75% GRK2 knock-down was achieved with the shRNA, compared to scramble shRNA control (Fig 5E). Treatment of the GRK2 shRNA chondrocytes with SAG showed normal accumulation of SMO in cilia of non-induced cells, in contrast to DOX-induced cells that failed to accumulate SMO in cilia (Fig 5F).

The effect of GRK2 removal, or inhibition of its kinase activity, on SMO accumulation in cilia varied between mouse and human cells. Mouse NIH3T3 $Grk2^{-/-}$ cells responded to SAG treatment with SMO accumulation in the cilia, while still failing to process Gli3 and induce Gli1 expression (Fig EV3A and B). Similarly, inhibition of GRK2 activity in wild-type ($Grk2^{+/+}$) NIH3T3 cells, by paroxetine or CMPD101, did not prevent SAG-induced SMO accumulation in the cilia (Fig EV3C and E). Similar data were found in IMCD3 mouse cells, where downregulation of GRK2 expression by DOX-inducible shRNA did not abolish SMO accumulation in the cilia (Fig EV3I and J). We also looked at the SAG-mediated SMO accumulation in cilia of NIH3T3 cells differentiated toward chondrocytes in micromass culture (Fig 3F). Surprisingly, the $Grk2^{-/-}$ NIH3T3 micromass cultures failed to accumulate SMO in cilia, while wild-type micromasses accumulated SMO in cilia normally (Fig EV3G and H). Regardless of SMO accumulation, all mouse model cell lines had impaired response to SAG, as shown by analyses of GLI3 processing and induction of GLI1 expression (Fig EV3B, D, F and H). Also, the lack of SMO phosphorylation was found in the $Grk2^{-/-}$ NIH3T3 cells (Fig EV3K), similar human $GRK2^{-/-}$ patient chondrocytes (Fig 4D).

## Loss of GRK2 impairs canonical Wnt signaling

Because GRKs such as GRK5/6 are known to phosphorylate the Wnt co-receptor LRP6 and thereby activate Wnt signaling (Chen *et al*, 2009), as Wnt signaling plays a key role in skeletogenesis (Gong *et al*, 2001; Laine *et al*, 2013), and as loss of GRK2 produced a profound skeletal disorder, we asked whether canonical (i.e., β-catenin dependent) Wnt signaling is impaired in $GRK2^{-/-}$ cells. Control and $GRK2^{-/-}$ fibroblasts were transfected with the TOPflash luciferase reporter for detection of Wnt/β-catenin transcriptional activity. Control cells responded to Wnt3A treatment with potent TOPflash *trans*-activation, in contrast to $GRK2^{-/-}$ cells in which the response to Wnt3A was markedly abrogated (Fig 6A). We next focused on the phosphorylation within the intracellular PPPS/TP motifs on LRP6. Ser/Thr phosphorylation in PPPS/TP motifs is a critical event in Wnt/β-catenin signaling, because it provides binding sites for AXIN1 and GSK3, leading to removal of the two

proteins from the β-catenin destruction complex, and its dissolution (Wolf *et al*, 2008). Defective TOPflash *trans*-activation in $GRK2^{-/-}$ cells correlated with reduced phosphorylation of LRP6 in response to Wnt3A, as measured with antibodies specific to two of the five phosphorylated PPPS/TP sites on LRP6, S1490, and T1572 (Fig 6B and C). In addition to the diminished phosphorylation of LRP6 in response to Wnt3A, we also detected decreased levels of total LRP6 in $GRK2^{-/-}$ cells (Fig 6C). Moreover, $GRK2^{-/-}$ cells showed increased steady-state phosphorylation of disheveled 2 (DVL2) (Fig 6B and C), an important signaling protein required for Wnt-induced LRP6 phosphorylation (Bilic *et al*, 2007), which may suggest that there is compensation for insufficient signal transduction through LRP6 phosphorylation in the absence of GRK2. A similar cellular phenotype has been previously observed in other mutants that affect signal transduction at the level of Wnt receptor complexes (Bryja *et al*, 2007). Together, these results demonstrated an impaired capacity of $GRK2^{-/-}$ cells to respond to canonical Wnt pathway activation.

Next, the mechanism by which GRK2 regulates Wnt signaling was interrogated. In 293T cells transfected with LRP6 and GRK2, no increased LRP6 phosphorylation was detected (unpublished observation), suggesting that GRK2 is not a LRP6 kinase. Next, we focused on ARRB2, which is well-known partner of GRK2, involved in GRK2-mediated agonist-induced desensitization b2-adrenergic receptors (Goodman *et al*, 1996). Importantly, ARRB2 is also a component of the canonical Wnt pathway, where it associates with the transmembrane Wnt signaling complex containing LRP6, FZD, and DVL. Importantly, the ARRB2 association with Wnt signaling complex is necessary for proper activation of pathway in response to Wnt ligands (Bryja *et al*, 2007). We asked whether ARRB2 interaction with FZD is affected by the GRK2 loss. The FZD4 and ARRB2 were expressed in R05-365A patient cells, and their interaction was probed by proximity ligation assay. The results show a loss of interaction of FZD4 with ARRB2 in the GRK2$^{-/-}$ patient cells, compared to control fibroblasts (Fig 6D).

To confirm the data obtained with $GRK2^{-/-}$ patient fibroblasts, we again used the NIH3T3 $Grk2^{-/-}$ cells (Pusapati *et al*, 2018). The Wnt3A-mediated *trans*-activation of TOPflash reporter was significantly downregulated in $Grk2^{-/-}$ NIH3T3 cells compared to wild-type controls (Fig 7A). This was accompanied by downregulated total LRP6 protein in $Grk2^{-/-}$ NIH3T3 cells, and under-phosphorylation of pLRP6$^{S1490}$ in response to Wnt3A (Fig 7B). In rat chondrosarcoma (RCS) chondrocytes and control human R00-082 chondrocytes, treatment with GRK2 inhibitor CPMD101 produced no effect on LRP6 protein expression, but inhibited the Wnt3A-mediated phosphorylation of LRP6 on T1572 and S1490 (Fig 7C and D). In T-REx™-293 cells, RCS chondrocytes, and control human R92-284 and R00-082 chondrocytes, inhibition of GRK2 kinase activity by paroxetine caused diminished response to Wnt3A in the TOPflash assay (Fig EV4A), downregulated total LRP6, and impaired LRP6 phosphorylation in response to Wnt3A (Fig EV4B–E). As GRK2 loss in the R05-365A patient fibroblasts and $Grk2^{-/-}$ NIH3T3 cells lead to downregulation of the LRP6 protein levels, we aimed to see whether the levels of the Frizzled (FZD) Wnt receptor were also affected in these cells. The Western blots showed that both cell types express FZD5, but not FZD4 or FZD10, and that the loss of GRK2 does not affect FZD5 expression (Fig EV5). Finally, $Grk2^{-/-}$ NIH3T3 cells showed a significant reduction of FZD4

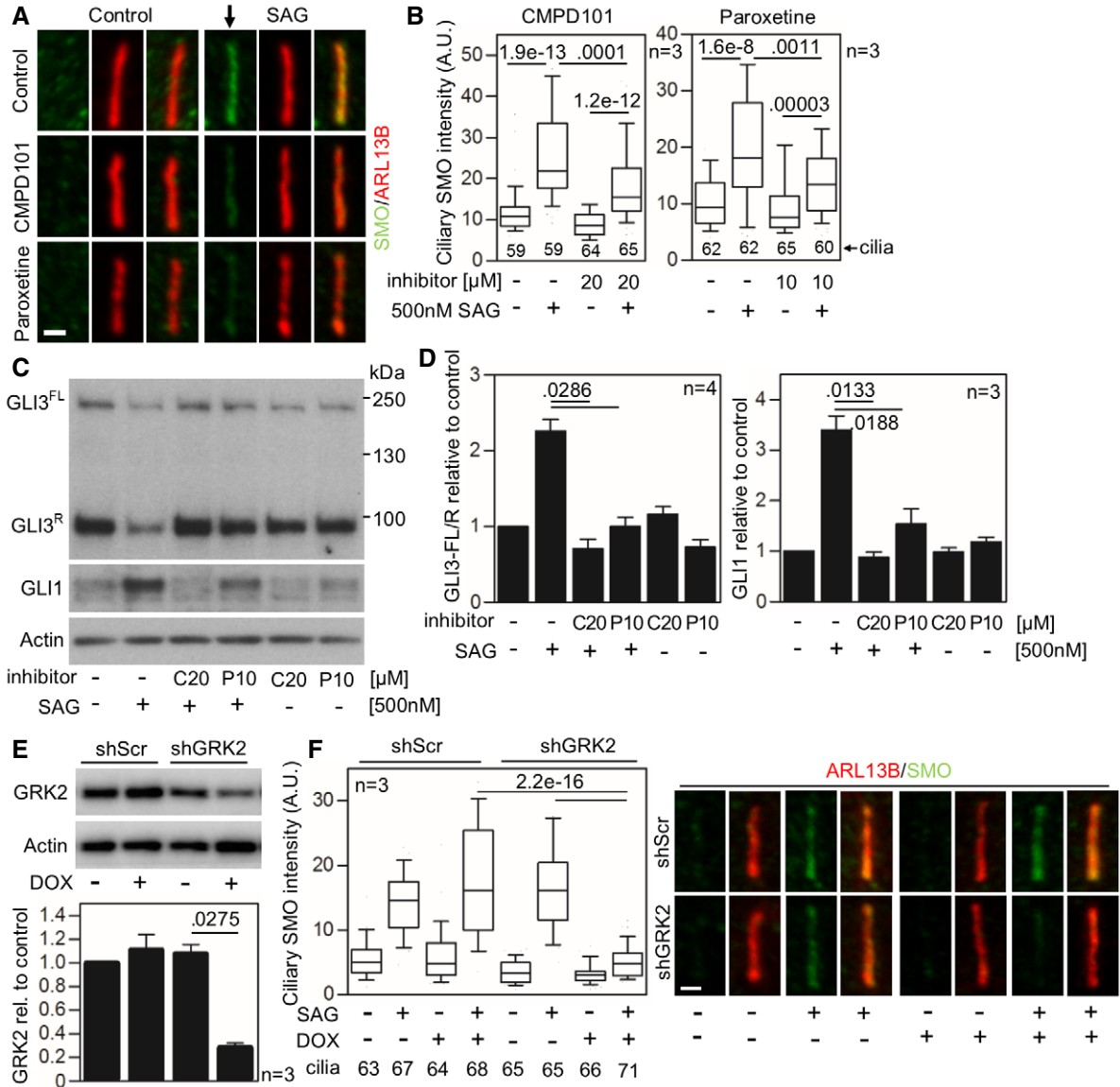

**Figure 5. Inhibition of GRK2 activity decreases Smoothened accumulation in cilia and GLI3 processing.**

A–D  Defective SMO cilia accumulation and inhibited Hh signaling in chondrocytes treated with GRK2 inhibitors. Control human R00-082 chondrocytes were serum starved in the presence of a GRK2 inhibitor, either CMPD101 (C) or paroxetine (P), for 24 h before they were treated with SAG for additional 12 (A, B) or 24 h (C, D). (A,B) Cilia were stained by ARL13B and SMO antibodies, and the intensity of ciliary SMO was analyzed and plotted. Note the impaired SMO accumulation in cilia caused by GRK2 inhibition (black arrow). Central band, median. Box, $1^{st}$-$3^{rd}$ quartile. Whiskers, 10%-90% percentile. Mann–Whitney $U$ test; number of biological experiments and the total numbers of analyzed cilia are indicated. Scale bar, 1 μm. (C) Cell lysates were immunoblotted for GLI3 processing and GLI1 upregulation; actin levels served as a loading control. $GLI3^{FL}$ and $GLI3^{R}$, full-length, and repressor GLI3 variants, respectively. (D) Quantification of GLI3 and GLI1 levels by densitometry. Note the impaired GLI3 processing and no GLI1 upregulation in cells treated with CMPD101 or paroxetine. Mean ± SEM. Mann–Whitney $U$-test (GLI3 ratio) and Welch's $t$-test (GLI1 levels); number of biological experiments is indicated.

E, F  Defective SMO cilia accumulation in chondrocytes with downregulated GRK2. (E) Doxycycline (DOX)-inducible GRK2 downregulation in control human R00-082 chondrocytes, tested by Western blot, normalized to actin, and plotted below. Mean ± SEM. Welch's $t$-test; number of biological experiments is indicated. shScr, scramble shRNA. (F) Cells pre-treated with DOX for three days were serum starved, SAG treated, stained, and analyzed as in (A,B), and the intensity of ciliary SMO was analyzed and plotted. Note the impaired SMO accumulation in cilia caused by GRK2 loss. Central band, median. Box, $1^{st}$-$3^{rd}$ quartile. Whiskers, 10%-90% percentile. Mann–Whitney $U$ test; number of biological experiments and the total numbers of analyzed cilia are indicated. Scale bar, 1 μm.

Source data are available online for this figure.

interaction with ARRB2, when compared to wild-type ($Grk2^{+/+}$) cells (Fig 7E), a similar finding as in GRK2-/- patient cells (Fig 6D). This phenotype was rescued by Grk2 add-back into the $Grk2^{-/-}$ NIH3T3 cells (Fig 7F), suggesting that one mechanism how GRK2 regulates Wnt signaling is promotion of ARRB2 association with the Wnt signaling complex.

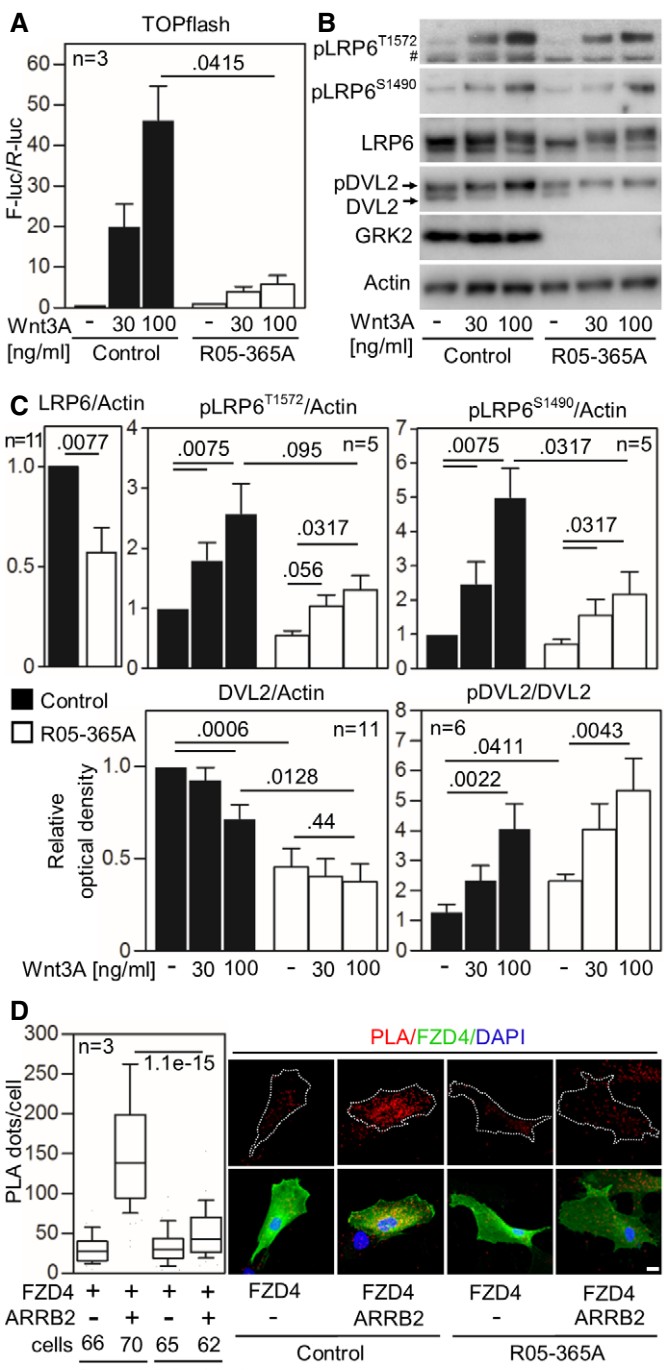

**Figure 6. Loss of GRK2 impairs canonical Wnt signaling.**

A    Control and $GRK2^{-/-}$ fibroblasts were transfected with the TOPflash Firefly (F) luciferase vector together with a control *Renilla* (R) luciferase vector, and the effect of Wnt3A on TOPflash transcriptional activation was determined by dual-luciferase assay. Note the impaired response to Wnt3A in $GRK2^{-/-}$ cells. Mean ± SEM. Welch's *t*-test; number of biological experiments is indicated.

B, C  Low phosphorylation (p) of Wnt3A co-receptor LRP6 as determined by Western blot with pS1490- and pT1572-LRP6 specific antibodies, reduced LRP6 and DVL2 expression, and increased steady state phosphorylation of DVL2. A representative experiment of four is shown. #, nonspecific band. (C) Densitometry of Western blot results demonstrating decreased levels of LRP6 in $GRK2^{-/-}$ cells, lower response of $GRK2^{-/-}$ cells to Wnt3A by LRP6 phosphorylation at S1490, and tendency toward the similar defect at T1572. Densitometry also showed less of total DVL2 levels, and increased steady state pDVL2 in $GRK2^{-/-}$ cells. Mean ± SEM. Mann–Whitney *U*-test; number of biological experiments is indicated.

D    Loss of interaction between Frizzled 4 (FZD4) and ß-Arrestin 2 (ARRB2) in $GRK2^{-/-}$ fibroblasts. Proximity ligation assay (PLA) between expressed FZD4-GFP and ARRB2-Flag showed significantly reduced proximity events between the two proteins in $GRK2^{-/-}$ cells. White dashed lines outline the transfected cells. Central band, median. Box, 1st–3rd quartile. Whiskers, 10–90% percentile. Mann–Whitney *U*-test; number of biological experiments and the total numbers of analyzed cells are indicated. Scale bar, 10 μm.

Source data are available online for this figure.

GRK2-inhibited human chondrocytes (Figs 4D–G and 5A, B, E, F, and EV2A and B), demonstrating that GRK2 activity is critical for SMO accumulation in human primary cilia. In addition, there was no transcriptional induction of Hh response genes in the $GRK2^{-/-}$ cells, so the cells were functionally nonresponsive to Hh signaling (Fig 4C). These data contrast with findings in mouse cells, in which the SMO accumulation was abrogated only at one case, i.e., in $Grk2^{-/-}$ NIH3T3 cells that were primed toward chondrocytes in micromass culture. In undifferentiated $Grk2^{-/-}$ NIH3T3 cells, in NIH3T3 cells treated with Grk2 inhibitors (Zhao *et al*, 2016; Pusapati *et al*, 2018) or in IMCD3 cells with downregulated Grk2, the SMO accumulated in cilia upon SAG normally, despite its under-phosphorylation (Fig EV3). This suggests that the SMO ciliary trafficking requires GRK2 in a cell context-dependent manner, and that, in human skin fibroblasts and cultured chondrocytes, this process is more dependent on the GRK2 activity. Notably, the SAG-mediated GLI3 activation and GLI1 induction were inhibited in all mouse and human cell models used in this study, suggesting the Hedgehog signaling is impaired in the absence of GRK2 or its activity, regardless of SMO ciliary accumulation (Zhao *et al*, 2016; Pusapati *et al*, 2018; preprint: Happ *et al*, 2020).

According to the current understanding, a massive SMO phosphorylation is necessary for its full activation and for the expression of high-threshold Hh genes, with the extent of the Hh pathway activity correlating with the SMO phosphorylation status (Jia *et al*, 2004; Chen *et al*, 2011). Specifically in vertebrates, Hh ligand binding to its receptor PTCH1 causes CK1α to phosphorylate the proximal regions of the C-terminal tail of SMO. This leads to a more open conformation of the SMO C-tail and prompts GRK2 to phosphorylate SMO at several more sites, leading to its full activation (Chen *et al*, 2004, 2010, 2011; Meloni *et al*, 2006; Zhao *et al*, 2007; Kovacs *et al*, 2008). Because CK1α phosphorylation sites on SMO partially overlap with those of GRK2 (Chen *et al*, 2011), it is unclear why, in the

# Discussion

## Loss of GRK2 results in ATD and loss of Hh signaling

We determined that loss of GRK2 in humans results in the ATD phenotype, characterized by marked skeletal abnormalities, an atrial septal defect and significant respiratory impairment. Loss of GRK2 resulted in under-phosphorylation of SMO and its exclusion from the cilia in cultured patient cells, and in GRK2-downregulated and

absence of GRK2, CK1α-mediated phosphorylation does not partially rescue the SMO signaling. Similarly, why other GRKs cannot phosphorylate SMO in the absence of GRK2 is surprising in that current thinking postulates that all GRKs are able to phosphorylate any of their GPCR targets, albeit with different efficiencies (Gurevich *et al*, 2012). Future studies will be needed to define the hierarchy of activity of the various molecules that phosphorylate SMO to mediate Hh

signal transduction. Interestingly, the distinct phenotype produced by loss of GRK2 suggests that GRK2 has distinct roles, particularly in skeletogenesis.

No gross CNS abnormalities were observed in the *GRK2* null individuals. This is surprising since the CNS patterning depends on Hh activity (Ribes & Briscoe, 2009), and the patient's *GRK2*⁻/⁻ skin fibroblasts did not respond to SAG stimulation. The published

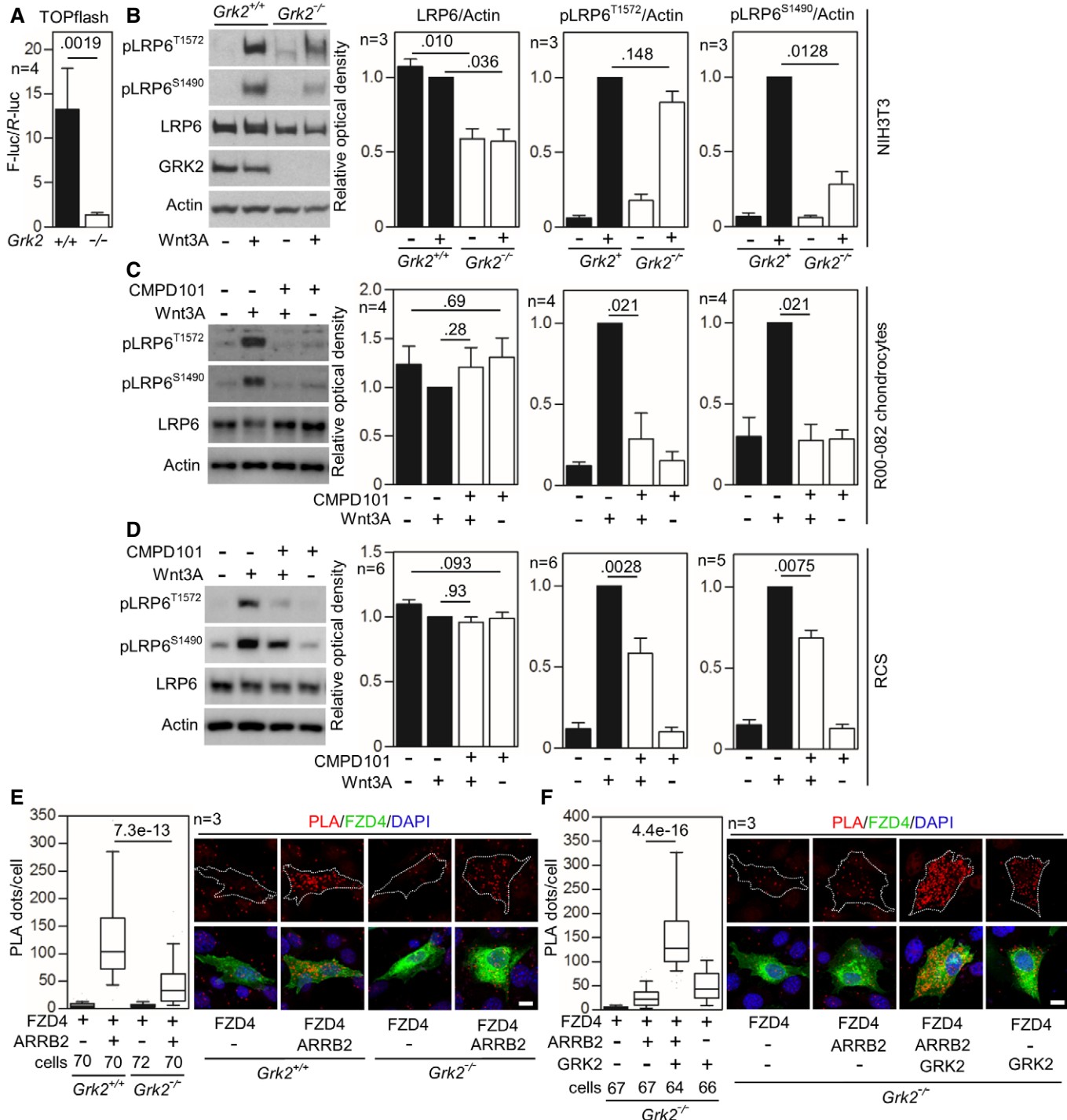

**Figure 7.**

**Figure 7. Inhibition of GRK2 activity inhibits canonical Wnt signaling.**

A   Control (+/+) and *Grk2*$^{-/-}$ NIH3T3 cells were transfected with the TOPflash Firefly (F) luciferase vector together with a control *Renilla* (R) luciferase vector, treated with Wnt3A (100 ng/ml) for 24 h and subjected to dual-luciferase assay. Note the impaired response to Wnt3A in *Grk2*$^{-/-}$ cells. Mean ± SEM. Mann–Whitney *U* test; number of biological experiments is indicated.

B   Control (+/+) and *Grk2*$^{-/-}$ NIH3T3 cells were treated with 40 ng/ml Wnt3A for 1 h. Note the lower LRP6 expression and inhibited Wnt3A-induced LRP6 phosphorylation in the *Grk2*$^{-/-}$ NIH3T3 cells. Mean ± SEM. Welch's *t*-test; number of biological experiments is indicated.

C, D   Loss of LRP6 phosphorylation in chondrocytes treated with the GRK2 inhibitor CMPD101. Control human R00-082 chondrocytes (C) and rat chondrosarcoma (RCS) cells (D) were treated with 20 μM CMPD101 overnight and then treated with 100 ng/ml Wnt3A for 1 h. Note the less LRP6 phosphorylation (pS1490- and pT1572-LRP6) in cells treated with CMPD101. Mean ± SEM. Mann–Whitney *U* test; number of biological experiments is indicated.

E, F   Loss of interaction between Frizzled 4 (FZD4) and ß-Arrestin 2 (ARRB2) in *Grk2*$^{-/-}$ NIH3T3 cells. (E) Proximity ligation assay (PLA) between expressed FZD4-GFP and ARRB2-Flag showed significantly reduced proximity events between the two proteins in *Grk2*$^{-/-}$ cells. (F) GRK2 add-back rescued the interaction. White dashed lines outline the transfected cells. Central band, median. Box, 1$^{st}$–3$^{rd}$ quartile. Whiskers, 10–90% percentile. Mann–Whitney *U* test; number of biological experiments and the total numbers of analyzed cells are indicated. Scale bars, 10 μm.

Source data are available online for this figure.

studies show that deletion of Grk2 in the mouse leads only to mild defects in the neural tube patterning (Philipp *et al*, 2008), and that loss of both Grk2 and Grk3 is required to fully inhibit the GLI3 processing and subsequently the Hh target gene expression in the murine neural precursor cells (Pusapati *et al*, 2018). This suggests at least partially redundant functions of Grk2 and Grk3 in cells and tissues expressing both proteins, and could explain the absence of CNS abnormalities in the *GRK2* null patients. Unlike neural precursors, the mouse embryonic fibroblasts and NIH3T3 cells do not express Grk3 (Zhao *et al*, 2016; Pusapati *et al*, 2018); therefore, genetic ablation of Grk2 alone is sufficient to obtain full inhibition of the Hh pathway in these cells. In line with these observations, the complete inhibition of the SAG-mediated GLI3 processing and *GLI1* and *PTCH1* expression in the *GRK2*$^{-/-}$ skin fibroblasts suggest they either do not express GRK3 or (less likely) that GRK3 does not contribute to Hh signaling in these cells. It is of note, however, that Grk3 knock-out mice develop normally (Peppel *et al*, 1997).

Most IFT mutants causing ATD and SRPS disrupt cilia number and architecture, which impairs Hh signaling as well as other ciliary functions. However, the cilia of *GRK2*$^{-/-}$ cells were structurally normal, suggesting that the phenotype is mediated through abnormal cilia function. In control cells, Hh pathway activation results in activation of SMO and degradation of GLI3$^R$, increasing the GLI3$^{FL}$/GLI3$^R$ ratio and inducing expression of downstream target genes (Wang *et al*, 2007). In *GRK2*$^{-/-}$ cells, the level of GLI3$^R$ was maintained and target gene expression could not be induced. Loss of Hh signaling and the resulting reduction of growth plate chondrocyte proliferation is likely to be mediated in part through defective or loss of Indian hedgehog (IHH) signaling. *Ihh* knock-out mice show severe dwarfism with abnormal axial and appendicular skeletal elements due to the marked reduction of chondrocyte proliferation and poorly organized, smaller hypertrophic zones (St-Jacques *et al*, 1999). IHH is produced by post-mitotic chondrocytes in the early stage of hypertrophic differentiation and regulates proliferation by increasing the expression of the chondrocyte mitogen PTHrP in the perichondrium (Kronenberg, 2006). The defective endochondral ossification resulting from loss of GRK2 is consistent with dependence of growth plate chondrocytes on IHH signaling.

## GRK2 in regulation of canonical Wnt signaling

Although there were similarities between the *Ihh*$^{-/-}$ mouse (St-Jacques *et al*, 1999) and *GRK2*$^{-/-}$ human growth plates, the latter exhibited a more marked loss of hypertrophic chondrocytes

(Fig 3B). This difference suggested that perhaps reduced Hh signaling was not the sole contributor to the cartilage growth plate pathology and mineralization defect. Based on the roles of paralogous GRKs in Wnt signaling, we tested the hypothesis that the human *GRK2*$^{-/-}$ cells might have defective Wnt signaling. The impaired transcriptional response to the canonical Wnt ligand, Wnt3A, in *GRK2*$^{-/-}$ cells, was accompanied by downregulated LRP6 expression and markedly diminished phosphorylation of PPPS/TP motifs within the LRP6 cytoplasmic tail, demonstrating that defective canonical Wnt signaling also contributes to the ATD phenotype.

In canonical Wnt signaling, the transcriptional activity of β-catenin is low due to its degradation mediated by the cytoplasmic destruction complex composed of Axin, Adenomatosis polyposis coli (APC), and GSK3. Canonical Wnt ligands including Wnt3A cause dissolution of the destruction complex leading to β-catenin stabilization and activation of β-catenin-dependent transcription (Nusse & Clevers, 2017). Phosphorylation at ser/thr residues in several PPPS/TP motifs within the cytoplasmic tail of Wnt co-receptor LRP6, mediated by GSK3, is critical events in the cellular response to canonical Wnts, because these motifs sequester GSK3 away from the destruction complex, efficiently eliminating its function (Cselenyi *et al*, 2008). Some GRKs such as GRK5/6 are also known to activate Wnt signaling via LRP6 phosphorylation (Chen *et al*, 2009). We found GRK2 unable to phosphorylate LRP6 in cells, suggesting that GRK2 is likely not a LRP6 kinase (unpublished observation). Instead, the loss of GRK2 impaired interaction between FZD4 and ARRB2 (Figs 6D and 7E). In canonical Wnt signaling, ARRB2 promotes formation of a ternary ARRB2-FZD-DVL complex, which promotes activation of the Wnt signaling through sequestration of axin away from the β-catenin destruction complex, leading to dissolution of destruction complex (Bryja *et al*, 2007; Rosanò *et al*, 2009). Recruitment of DVL to FZD depends on phosphorylation of the FZD C-terminus by kinases, including CK1 and GRK2 (Strakova *et al*, 2018). It is therefore possible that the inhibited response of GRK2-null cells to Wnt stimulation is due to abolished formation of the FZD-DVL-ARRRB2-axin complex.

Our findings differ from the findings of Wang *et al* (2009) that indicate that GRK2 negatively regulates Wnt signaling through interactions between GRK2 and APC protein. Their findings suggest that loss of GRK2 would have a positive effect on bone mineralization, in contrast to our findings of a negative impact on bone mineralization based on the radiographic phenotype.

Canonical Wnt signaling is a critical regulator of chondrocyte differentiation and promotes the development of hypertrophic chondrocytes via inhibition of mitogenic PTHrP signaling and induction of hypertrophic chondrocyte proteins such as type X collagen and matrix metalloproteinase 13 (Tamamura *et al*, 2005; Dong *et al*, 2006; Guo *et al*, 2009). In mice, constitutive activation of β-catenin-mediated transcription promoted chondrocyte hypertrophy and maturation, leading to an elongated hypertrophic zone. In contrast, β-catenin inactivation resulted in small hypertrophic zones, disorganized pre-hypertrophic chondrocytes, and delayed primary ossification (Dao *et al*, 2012). The similarity of the latter phenotype with the shortened and disorganized hypertrophic cartilage, and poorly developed primary bone observed in the growth plate suggests that the loss of GRK2 negatively affects skeletogenesis, in part mediated by defective canonical Wnt signaling.

### GRK2 in skeletogenesis

Loss of GRK2 in humans produces a severe, but not an early embryonic lethal skeletal disorder. By contrast, loss of GRK2 in mice ($Grk2^{-/-}$ or $\beta ark1^{-/-}$) resulted in embryonic lethality between E9 and E15.5 (Jaber *et al*, 1996). The $Grk2^{-/-}$ embryos were described as smaller than their wild-type and heterozygous littermates and had significant hypoplasia of the ventricular myocardium (Jaber *et al*, 1996). Similarly, conditional *Grk2* mice with *EIIA-Cre*-mediated germline deletion of *Grk2* exhibited early embryonic lethality at E10.5; embryos were also described as smaller and having hypoplasia of a single ventricle, suggesting impaired or delayed heart development (Matkovich *et al*, 2006). Conditional ablation of *Grk2* in cardiomyocytes using *Nkx2.5-Cre* showed normal cardiac development and viability, but the mice did show enhanced baseline contractility and diminished β-adrenergic receptor-mediated tachyphylaxis (Matkovich *et al*, 2006). Thus, the mouse data demonstrated that overall growth and heart development in mice depend on GRK2. One patient had a cardiac defect, but it has not been previously appreciated that GRK2 has a profound effect on skeletogenesis.

In summary, this study demonstrates that loss of GRK2 causes ATD via defects in ciliary signaling rather than as a consequence of a structural defect in primary cilia. We confirmed that GRK2 is a positive regulator of Hh signaling, and that its absence results in inhibition of the GLI3 activity. We also show that loss of GRK2 activity results in defective ciliary accumulation of SMO in human fibroblasts and chondrocytes, and in chondrocytic micromasses produced from murine NIH3T3 cells. We also demonstrated impaired canonical Wnt signaling in $GRK2^{-/-}$ cells and showed that this defect stems from reduced expression and phosphorylation of LRP6 in response to canonical Wnt ligands. Finally, a severe under-sulfation of extracellular matrix proteoglycans was found in the ATD cartilage, and in the cartilaginous micromass cultures generated from Grk2-null or Grk2-inhibited cells (Fig 3D–F), suggesting that GRK2 regulates proteoglycan metabolism. The maintenance of proteoglycan homeostasis is important for the cartilage growth, as numerous skeletal dysplasias associate with deficient sulfation (Paganini *et al*, 2020). Apart from organization of the extracellular matrix network, the sulfated proteoglycans also promote the Ihh and TGFβ signaling, which is essential for chondrocyte proliferation and differentiation (Klüppel *et al*, 2005; Cortes *et al*, 2009; Gualeni *et al*, 2010). The mechanism of how GRK2 regulates proteoglycan sulfation is not known, and is opened for future research.

Further investigations should reveal which of the many potential targets of GRK2 are contributing to its role in ATD. Apart from *GRK2*, mutations in two other ser/thr kinases, *ICK* and *NEK1*, are also associated with skeletal ciliopathies (Thiel *et al*, 2011; Paige Taylor *et al*, 2016), together revealing an emerging theme that regulation of cell function by ser/thr kinases plays a role in cilia disorders, as well as in skeletal patterning and linear growth.

## Materials and Methods

### Ascertainment

Affected individuals were ascertained under human subjects' protocols, and the diagnosis of ATD was made by reviewing clinical records and radiographic images. Each proband was assigned a reference number, R05-365A and Cmh001543-01. In the case of R05-365A, a skin fibroblast culture was established and a section of distal femur was placed in paraffin for histologic analysis. Human subject approval for the affected individual and their unaffected family members was obtained through an approved University of California at Los Angeles human subject protocol that follows principles set forth in the WMA Declaration of Helsinki and the Department of Health and Human Services Belmont Report. Human data and samples can be obtained with proper institutional human subjects approval.

### Exome analysis

Genomic DNA was isolated from cultured fibroblasts and serum using a kit, according the manufacturer's protocol (Qiagen). Library construction and exome sequencing of DNA from the proband was carried out at the University of Washington Center for Mendelian Genomics and Children's Mercy, Center for Pediatric Genomic Medicine. The exome sequencing libraries were prepared with the NimbleGen SeqCap EZ Exome Library v2.0 kit and sequenced on the Illumina GAIIx platform. Reads were mapped with Novoalign and variants called using the Genome Analysis Toolkit following their Best Practices recommendations. Variants were filtered against public databases and annotated as previously described (Lee *et al*, 2012). The identified variants in *GRK2* were confirmed by Sanger sequence analysis of amplified DNA.

### Animal experiments

ISA brown fertilized chicken eggs were obtained from Integra farm (Zabcice, Czech Republic). Chicken embryos were selected randomly for the study. All fertilized eggs without developmental defects were used for the injection. Eggs were incubated in a humidified incubator at 37.8°C. CMPD101 (10 mM; Tocris) was applied into the right wing bud at stage HH20-22. Micromanipulator (Leica) and a microinjector (Eppendorf) were used to better target the selected area of application. Embryos were collected after 10–12 days of incubation following the treatment. Both right (injected) and left (untreated) wings were fixed in 100% ethanol, stained with Alizarin red/Alcian blue solution, and cleared in KOH/glycerol. Injection of CMPD was performed in 2 independent experiments: 17

embryos, 42 embryos; 18 embryos died shortly after injection, and the skeletal analysis was performed on 27 embryos. Bones that could not be accurately measured due to insufficient quality of the acquired pictures or the bone preparation were excluded from the analysis. Micromasses were established from chicken embryos at 3 independent experiments from approximately 80 embryos for each experiment. Mesenchymal cultures were established from the anterior and posterior parts of forelimb buds of stage HH20-21 chicken embryos. Wing buds were dissected into Pucks saline A (0.8% NaCl, 0.04% KCl, 0.035% NaHCO$_3$, 0.1% glucose), digested by Dispase II (1 U/ml; Sigma), and filtered to obtain a single cell population. Cells were resuspended at a density of $2 \times 10^7$ cells/ml and plated in 10 µl aliquots in 6-well culture dishes. Cells were left to adhere for 1 h in the incubator before 2 ml of culture media (F12/DMEM supplemented with 10% FBS, penicillin, streptomycin, glutamine, 1% ascorbic acid, and 10 mM β-glycerol phosphate) containing CMPD101 was added to each well. Micromasses were cultured for 5 days. For Alcian blue staining, micromass cultures were fixed by 4% paraformaldehyde, washed in PBS, and stained overnight with 1% Alcian blue in 0.1 M HCl (pH1.0) or 1% Alcian blue in 3% CH$_3$COOH (pH2.5). Images were taken with a Leica S6D (Leica) microscope, and signal was quantified in Adobe Photoshop 7.0. The analyses were done blinded by a researcher different from the one doing injection. All procedures were performed according to the experimental protocols and rules established by the Laboratory Animal Science Committee of the Institute of Animal Physiology and Genetics (Libechov, Czech Republic).

## Cell lines, transfection, micromass culture, shRNA, luciferase reporter assay, and RT–PCR

Control and R05-365A skin fibroblasts (38 weeks gestation), and R00-082 (22 weeks), and R92-284 (22 weeks) control chondrocytes were obtained under an approved University of California at Los Angeles human subjects protocol. Primary cultures of cells isolated from skin or growth plate cartilage were cryopreserved in liquid nitrogen; low passage number cells (<p. 6) were used in the experiments. T-REx™-293 cells were purchased from Thermo Scientific. RCS cells were obtained from B. de Crombrugghe (Mukhopadhyay *et al*, 1995). Grk2$^{-/-}$ NIH3T3 cells were previously described (Pusapati *et al*, 2018). Cells were propagated in DMEM media supplemented with 10% FBS and antibiotics (Invitrogen), and starved in 0.5% serum (chondrocytes and human fibroblasts) or in 0.1% FBS (NIH3T3). IMCD3 were obtained from ATCC, propagated in DMEM:F12 (1:1) supplemented with 10% FBS and antibiotics, and starved without serum. All cell lines were routinely tested for mycoplasma infection using DAPI staining. For micromasses, the NIH3T3 cells were seeded in 10 µl aliquots in 6-well plate at density $1 \times 10^7$ cells/ml and left to adhere for 2 h before 2 ml of culture media (F12/DMEM supplemented with 10% FBS, penicillin, streptomycin, glutamine, 1% ascorbic acid, and 10 mM β-glycerol phosphate) containing CMPD101 was added to each well. Micromasses were cultured for 7 days, and the medium, supplemented with CMPD101, was changed every other day. For shRNA, the murine IMCD3 and human R00-082 cell lines with stably integrated, doxycycline-inducible expression of shRNA targeting GRK2 and the respective scramble controls were generated by lentiviruses. Lentiviral vector containing doxycycline-inducible U6 promoter and TetRep-P2A-Puro-P2A-mCherry (Eshtad *et al*, 2016;

Kunova Bosakova *et al*, 2019) (kindly provided by Mikael Altun) was modified to express shRNA by introducing the following oligonucleotides (mouse-shGRK2 forward: CCGGGGAGATCTTTGACTCCTATA TTCTCGAGAATATAGGAGTCAAAGATCTCTTTTTG, mouse-shGRK2 reverse: AATTCAAAAAGAGATCTTTGACTCCTATATTCTCGAGAAT ATAGGAGTCAAAGATCTC; mouse-scrambled forward: CCGGGATGC TAACGTCTTATACTTTCTCGAGAAAGTATAAGACGTTAGCATCTTT TTG, mouse-scrambled reverse: AATTCAAAAAGATGCTAACGTCT TATACTTTCTCGAGAAAGTATAAGACGTTAGCATC; human-shGRK2 forward: CCGGGAGGCTGACATGCGCTTCTATCTCGAGATAGAAGCG CATGTCAGCCTCTTTTTG, human-shGRK2 reverse: AATTCAAAAAG AGGCTGACATGCGCTTCTATCTCGAGATAGAAGCGCATGTCAGCC TC; human-scrambled forward: CCGGGGATTCAGCTGCACGACTTGT CTCGAGACAAGTCGTGCAGCTGAATCCTTTTTG, human-scrambled reverse: AATTCAAAAAGGATTCAGCTGCACGACTTGTCTCGAGACA AGTCGTGCAGCTGAATCC; cloned shRNA sequences were verified by Sanger sequencing. Lentiviral particles were generated as described previously (Peskova *et al*, 2019; Barta *et al*, 2016) using pMD2.G (Addgene #12259) and psPAX2 (Addgene #12260) (gift from Didier Trono). After transduction, at least $2 \times 10^4$ mCherry-positive cells were sorted using BD FACSAria™ II (BD Biosciences). Generated cell lines were propagated in the presence of 1 µg/ml puromycin. shRNA expression was induced by 1 µg/ml doxycycline (Invitrogen) for in total four days before the analyses. Wnt3A was obtained from RnD Systems, SAG from EMD Millipore, CMPD101, and paroxetine from Tocris. Cells were transfected by electroporation using the Neon Transfection System (Invitrogen). For the luciferase reporter assays, the µg ratio between the TOPflash luciferase vector (obtained from R. Moon) and the pRL-TK (Promega) control *Renilla* vector was 3:1. The luciferase activity was determined using a Dual-Luciferase Reporter Assay (Promega). For RT–PCR, total RNA was isolated using the RNeasy Mini Kit (Qiagen) and reverse transcribed with the First Strand cDNA Synthesis Kit (Roche). The following primers were used for RT–PCR: GRK2-F 5′-AGAGTGCCCACTGAGCATGTC-3′ (exon 5), GRK2-R 5′-CCTGC TTCATCTTGATGCGC-3′ (exon 9); GAPDH-F 5′-AGCCACATCGCT CAGACACC-3′, GAPDH-R 5′-GTACTCAGCGCCAGCATCG-3′. For qPCR, the following QuantiTect primers (Qiagen) were used: Hs_PTCH1_1_SG (QT00075824), Hs_GLI1_1_SG (QT00060501), and Hs_GAPDH_vb.1_SG (QT02504278).

## Western blots

Cells were extracted directly in Laemmli sample buffer (125 mM Tris–HCl pH 6.8, 4% SDS, 5% ß-mercaptoethanol, 10% glycerol, 0.01% bromphenol blue). Lysates were resolved by SDS–PAGE, transferred onto a PVDF membrane, and visualized by chemiluminescence (Thermo Scientific). For the phosphoshift PAGE, cells were transfected with the pEGFP-mSMO vector (#25395; Addgene) for 24 h, and lysed on ice for 30 min in RIPA buffer (50 mM Tris–HCl pH7.5, 150 mM NaCl, 5 mM EDTA, 1% IGEPAL CA-630) with freshly added 1 mM Na$_3$VO$_4$ and protease inhibitors (Roche). The samples were subjected to three freeze/thaw cycles before being loaded on the phosphoshift PAGE gels (6% acrylamide:bis-acrylamide, 99:1), and the electrophoresis run at constant 100V for 2 h. The following antibodies were used: LRP6 (3395), LRP6$^{S1490}$ (2568), DVL2 (3216), GLI1 (2643), and actin (3700) (Cell Signaling); actin (sc-1615, sc-1616) and GRK2 (sc-562) (Santa Cruz); GRK2 (G0296)

(Sigma); LRP6$^{T1572}$ (72187) and FZD5 (06-756; Millipore); GLI3 (AF3690) (RnD Systems); FZD4 (ab83042) and FZD10 (ab83044; Abcam); GFP (50430-2-AP, Proteintech); the final dilution was 1:1,000. Band intensities were quantified using ImageJ (http://www.imagej.nih.gov/ij/). In some in-cell experiment, the researcher analyzing the samples was not informed about the identity of the samples, or the idea of the experiment.

## Proximity ligation assay (PLA), immunochemistry, and microscopy

For the PLA, the cells were transfected with these vectors: FZD4-EGFP (a kind gift from Gunnar Schulte; Bryja *et al*, 2007), pcDNA3-Flag-ARRB2 and bGRK2 (kind gifs from Robert J. Lefkowitz; Strakova *et al*, 2018) in a 1:3:3 ratio, fixed 24 h later in paraformaldehyde, permeabilized in 0.1% Triton X-100, and stained by Duolink® PLA (Sigma) according to manufacturer's protocol. Rabbit FLAG (1:200; F7425; Sigma) and mouse GFP (1:200; sc-53882; Santa Cruz) antibodies were used for PLA; the GFP signal was used to identify the co-transfected cells. PLA counting analysis was done in Fiji (http://fiji.sc/Fiji). For immunocytochemistry, cells were fixed in paraformaldehyde and incubated with the following antibodies: acetylated α-tubulin (1:100; 32-2700; Invitrogen), ARL13B (1:300; 17711-1-AP; Proteintech), pericentrin (1:1,000; ab4448; Abcam), γ-tubulin (1:150; 66320-1-Ig; Proteintech), GLI3 (1:50; AF3690; RnD Systems), Smoothened (1:50; sc-166685; Santa Cruz), IFT43 (1:100; sc-245285; Santa Cruz), IFT81 (1:100; 11744-1-AP; Proteintech), IFT88 (1:100; 13967-1-AP; Proteintech), KIF3A (1:100; ab11259; Abcam), TRAF3IP (1:100; PA5-30507; Pierce), WDR34 (1:100; HPA041091; Sigma), and ICK (1:100; HPA001113; Sigma). Secondary antibodies conjugated with AlexaFluor488/594 (1:500; A21202, A21203, A21206, A21207, A11055) were from Invitrogen; the donkey anti-mouse AlexaFluor405 antibody was from Abcam (1:250; ab175658). Images were taken on a Carl Zeiss LSM700 laser scanning microscope with acquisition using ZEN Black 2012 software. Images were acquired using a 63× oil immersion objective as Z-stacks with 0.3 μm distance between neighboring z-sections. Measurements of cilia length in 3D were performed as previously described (Paige Taylor *et al*, 2016). The intensity of ciliary SMO was analyzed as previously described (Kunova Bosakova *et al*, 2018). For histology, growth plate samples were fixed with 4% PFA overnight and decalcified for 3 days. After decalcification, samples were embedded in paraffin and 10-μm sections were prepared. Samples were stained with Picrosirius red for 1 h and hematoxylin for contrast. For immunohistochemistry, sections were treated with citrate buffer for antigen retrieval and quenched using peroxidase solution, both for 10 min each. GRK2 antibody was obtained from Cell Signaling (3982; 1:100). Immunostained sections were developed with the Histostain Plus kit with DAB as the chromogen (Invitrogen). The purified rabbit IgG isotype standard (BD Biosciences) was used as a negative control. Alcian blue staining in the patient growth plates was performed using alcian blue/hematoxylin for 40 min and then processed through a series of washes. Brightness, contrast, and threshold were adjusted in some images for a clear data presentation, always equally through the entire image and the panel of images. Only raw images and blot scans without any adjustments were used for quantitative analyses. In some in-cell experiment, the researcher analyzing the samples was not informed about the identity of the samples, or the idea of the experiment.

---

### The paper explained

**Problem**
Herein, we identified patients with Jeune syndrome or asphyxiating thoracic dystrophy (ATD), a genetically heterogeneous disorder that is both multisystemic disorder and often associated with lethality. In two independent families, we identified homozygosity for a loss-of-function variant and biallelic variants predicting loss of functional protein. Because of this novel finding that links GRK2 to the skeletal development, we focused on identification of the molecular mechanisms underlying this disorder since loss of *Grk2* in mice is an early embryonic lethal.

**Results**
We found that the loss of GRK2 leads to specific changes in the bone that indicated impaired function of two major regulators of bone development, both Hedgehog and Wnt signaling. We indeed found that loss of GRK2 in patient's cells and model cell lines led to deregulation of these two pathways, suggesting in part the molecular mechanisms underlying this phenotype.

**Impact**
Development skeletal disorders, including ATD, are often severe, lethal syndromes with no cure or treatment options. Identification of the molecular pathogenesis of the disease therefore expands our understanding of the genetic heterogeneity associated with this disorder, provides families with reproductive options, and uncovers the role of GRK2 in skeletogenesis.

---

### Sample size and statistical analyses

The minimal number of independent biological experiments was set to three. Whenever applicable, technical replicates were done and eventually averaged for every biological experiment. With assays having higher variability, additional biological experiments were undertaken in order to present a clear information. A non-parametric Mann–Whitney *U*-test (for $n \geq 4$) and Welch's *t*-test ($n = 3$) were used to calculate the *P* values that are specified directly in the figure panels. Every figure and figure legends contain information on the sample size, the statistical test used, and the *P* value.

## Data availability

The numerical data, uncropped blots, and confocal images for all main figures and EV figures are available on the Journal's web page. This study includes no data deposited in external repositories.

**Expanded View** for this article is available online.

### Acknowledgements
This work was supported by NIH grants RO1 AR066124, R01 AR062651, and RO1 DE019567 to D.K and D.H.C. R.R. was supported by grant NIH/NIGMS GM118082. Sequencing was provided by the University of Washington Center for Mendelian Genomics (UW-CMG) and was funded by NHGRI and NHLBI grants UM1 HG006493 and U24 HG008956. The content is solely the responsibility of the authors and does not necessarily represent the official views of the National Institutes of Health. The NIH Training Grant in Genomic Analysis and Interpretation, T32 HG002536 supported S.P.T. This work was also

supported by Ministry of Education, Youth and Sports of the Czech Republic (KONTAKT II LH15231; LTAUSA19030); Agency for Healthcare Research of the Czech Republic (NV18-08-00567) and Czech Science Foundation (GA17-09525S, GA19-20123S) grants to P.K. Czech Science Foundation (GA18-17658S, GA17-16680S) supported V.B. Czech Science Foundation (16-24043J) supported J.K. A Junior Researcher Award from the Faculty of Medicine, Masaryk University supported M.Bo. I.D. is a Geisman Fellow supported by the Osteogenesis Imperfecta Foundation (OIF). The funders had no role in study design, data collection and analysis, decision to publish, or preparation of the manuscript. M.Bu. was supported by the project EXCELLENCE in molecular aspects of the early development of vertebrates from the Operational Programme Research, Development and Education (CZ.02.1.01/0.0/0.0/15_003/0000460), funded by the Ministry of Education, Youth and Sports of the Czech Republic. A.N. is a Brno Ph.D. Talent Scholarship Holder—Funded by the Brno City Municipality.

## Author contributions

MBo, SPT, DHC, PK, and DK designed the research; SPT collected the R05-365A patient data and analyzed primary cilia; ID and JM analyzed the growth plates; EH and MBu produced the animal and micromass experiments; TB produced the lentiviruses and shRNA cell lines; ETR and IT provided the Cmh001543-01 patient data and contributed to the exome analyses; MBo, SPA, AN, KS, MV, LB, JK, TR, GVP, VB, DAN, MJB, and RR performed the cell experiments; MBo, DHC, PK, and DK wrote the paper.

## Conflict of interest

The authors declare that they have no conflict of interest.

## For more information

- https://www.omim.org/entry/109635
- https://www.omim.org/entry/208500
- https://www.uclahealth.org/ortho/isdr

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
