## [Review Process File · EMBO Molecular Medicine]

Mutations in GRK2 cause Jeune syndrome by impairing Hedgehog and canonical Wnt signaling

Michaela Bosakova, Sara Abraham, Alexandru Nita, Eva Hrubá, Marcela Buchtová, Paige Taylor, Ivan Duran, Jorge Martín, Katerina Svozilová, Tomas Barta, Miroslav Varecha, Lukas Balek, Jiri Kohoutek, Tomasz Radaszkiewicz, Ganesh Pusapati, Vitezslav Bryja, Eric Rush, Isabelle Thiffault, Deborah Nickerson, Michael J Bamshad, Rajat Rohatgi, Daniel Cohn, Deborah Krakow, and Pavel Krejci
DOI: 10.15252/emmm.201911739

Corresponding author(s): Pavel Krejci (krejci@med.muni.cz), Deborah Krakow (dkrakow@mednet.ucla.edu), Pavel Krejci (krejci@med.muni.cz)

Review Timeline:

Submission Date:	10th Nov 19
Editorial Decision:	17th Dec 19
Revision Received:	30th Jul 20
Editorial Decision:	17th Aug 20
Revision Received:	8th Sep 20
Accepted:	15th Sep 20

Editor: Jingyi Hou

Transaction Report:

Thank you for the submission of your manuscript to EMBO Molecular Medicine. We have now received feedback from the three reviewers who agreed to evaluate your manuscript. As you will see from the reports below, the referees acknowledge the interest of the study. However, they also raise substantial concerns on your work, which should be convincingly addressed in a major revision of the present manuscript. In particular, it will be important to study the defects in Hh and/or Wnt signaling using patient samples (as suggested by reviewer #2), and to provide more mechanistic insights into the failed ciliary SMO localization in human GRK2 $-/-$ cells (as commented by reviewer #1). Moreover, attention should be given to placing the study in the context of existing literature.

Addressing the reviewers' concerns in full will be necessary for further considering the manuscript in our journal, and acceptance of the manuscript will entail a second round of review. EMBO Molecular Medicine encourages a single round of revision only and therefore, acceptance or rejection of the manuscript will depend on the completeness of your responses included in the next, final version of the manuscript. For this reason, and to save you from any frustrations in the end, I would strongly advise against returning an incomplete revision.

When submitting your revised manuscript, please carefully review the instructions that follow below.

Referee #1 (Remarks for Author):

Bosakova et al report the identification of two individuals exhibiting symptoms of asphyxiating thoracic dystrophy (ATD) or Jeune 18 syndrome, both of whom lack wild type GRK2 activity, due to homozygosity or compound heterozygosity for different mutant alleles of the GRK2 gene. The authors present data showing a complete loss of GRK2 mRNA and protein in fibroblasts derived from the individual homozygous for a premature termination codon allele, indicating this to be a null allele.

Histological analysis of femoral growth plate cartilage in this individual reveals some similarity to the defects seen in Indian Hedgehog (Ihh) mutant mice, consistent with the known role of GRK2 in Hh signal transduction. Surprisingly, the authors make no mention of the prior analysis of GRK2 in Hh signaling at this point, presenting their data as demonstrating "loss of GRK2 compromises HH signaling in cells, rendering them unresponsive to HH pathway activation" as though this were a novel finding (page 8 lines 11/12). This is extremely misleading and unfair to previous authors: the prior studies of GRK2 in Hh signalling should be referred to on page 7 as the basis for investigating disruption of Hh signaling in the patient-derived cells.

The main finding of this study is that loss of GRK2 activity in human cells inhibits the localisation of SMO to the primary cilium (PC) in response to the SMO agonist SAG. This is a surprising finding, given previous findings (confirmed by the authors) that GRK2 activity is not required for Smo localisation in murine cells. This conclusion is supported by experiments in which wild-type chondrocytes treated with GRK2 inhibitors also showed an impairment of SMO localisation to the PC. However, this effect was not absolute: in this regard, Figure 5a is somewhat misleading as it shows a complete absence of SMO in the PC of treated cells - there is no information about the concentration of inhibitor used on this particular sample - and Figure 5b shows that even at the highest dose, SMO still accumulates in about 35% of cases. It would also be of interest to know if the levels of SMO accumulation are altered in these cells, as appear to be the case for the NIH3T3 cells treated with the same inhibitors (Fig S2 C,E). In any event, these data in themselves do not support the statement made in the Discussion (Page 10 line 29) "that GRK-phosphorylation of SMO is required to activate the pathway". Such a conclusion requires that the authors investigate the phosphorylation of SMO in their mutant cells, something they do not report.

The authors also present evidence that loss of GRK2 compromises the response of cells to WNT signaling; in particular they describe a reduction in the levels of phosphorylated LRP6 (page 9 line 28); however, this appears to reflect an overall reduction in LRP levels and does not support the statement (page 12 line 25/26) that "GRK2 can also activate Wnt signaling via LRP6 phosphorylation).

In conclusion, this paper provides data consistent with, but not provide definitive proof of, a causal relationship between loss of GRK2 and ATD. A striking finding is the inhibition of SMO localization to the PC in patient derived cells. Uncovering the basis of this difference would be of great interest but the authors simply comment that its basis "remains unclear" (page 8 line 29). Without further investigation of this difference, it is questionable whether this report is of sufficient interest to merit publication in EMBO Molecular Medicine. Another important issue that is not raised is why the loss of SMO activity caused by the loss of GRK2 activity does not have more profound developmental: loss of HH signaling, for instance, is well known to cause major defects in CNS patterning, resulting in holoprosencephaly. The authors do hint at a possible specific role for GRK2 in skeletogenesis (page 11 line 21) but this is not explored further.

Referee #2 (Comments on Novelty/Model System for Author):

The major limitation of this study is the absence of an animal model to support the loss of GRK2 as the cause of skeletal phenotype in the patients. This is because complete loss of GRK2 is embryonic lethal in both mice and zebrafish. Theoretically, investigators could try to accomplish a conditional mouse model (with expression limited to cartilage) or studying cartilage development in zebrafish embryos, etc. However, to be practical, these models may not necessarily be adequate (for example a conditional mouse model could be early lethal as well) and I do appreciate the strong in vitro data that the authors included to support their conclusion.

Referee #2 (Remarks for Author):

Summary

The paper reports two families with asphyxiating thoracic dystrophy (ATD) and pathogenic variants in GRK2. A thorough clinical description, including radiographs and a detailed clinical table, supports the patients' phenotype to be consistent with a skeletal ciliopathy, specifically ATD. The investigators performed in vitro mechanistic studies using patient fibroblasts and cell lines to show that disruption of Hedgehog signaling (via impaired translocation of smoothed to the cilia) and abnormal canonical Wnt signaling (due to decreased levels and reduced phosphorylation of LRP6) contribute to the phenotype in GRK2-null cells. The manuscript contains novel data that is clinically relevant and of broad significance by describing a new form of skeletal dysplasia that is mediated by loss-of-function of GRK2, a ser/thr kinase. There are however a few issues that should be addressed by the authors.

Major Comments

1. The major weakness of this study is the lack of in vivo data to further support that loss of GRK2 is causing the skeletal phenotype in the patients. I realize this is difficult to overcome since GRK2^{-/-} mice and zebrafish show early embryonic lethality (necessitating a conditional model or studies in early embryonic stage). The authors may consider citing in the discussion previous paper showing severe growth retardation and abnormal limb development in the knockout mouse model (PMID 18815277).
2. As an alternative to animal model, and since cartilage growth plate is available from one of the subjects, the authors may consider studying the defect in Hh and/or wnt signaling in the affected tissue (for example by immunohistochemistry for components of the signaling pathway or known downstream targets).

Minor comments

1. The detailed clinical description is much appreciated. I suggest to edit and make the styling more consistent between the paragraphs of two families. Also, authors may consider replacing "case" (in abstract) with "proband" or "affected individual" similar to what they used later in the text.
2. It will be helpful to uniformly describe the variants using HGVS nomenclature, including hg19 genomic position, and authors may consider replacing the term "mutation" with "variant" or "pathogenic variant".

Referee #3 (Remarks for Author):

General:

This is well written manuscript describing a novel, ATD-like human phenotype due to biallelic GRK2 loss of function mutations. This will be of interest to a broad community of developmental biologists as well as human geneticists. The genetic data is convincing and functional data also of good quality.

The authors suggest the phenotype observed is JATD and that GRK2 loss of function causes a ciliopathy. The genetic data is convincing that the patient phenotype is a result of GRK2 loss of function. The phenotype resembles ATD / skeletal ciliopathies with regards to short ribs and long bones, however doesn't seem to be identical (eg no typical pelvis configuration, no polydactyly). Short ribs occur in different human disease and are not limited to ciliopathies eg can be observed in thanophoric dysplasia (FGFR3 variants) or Schachman-Diamond S. (SBDS variants). I would therefore suggest to label the phenotype an ATD-like hedgehog-related chondrodysplasia or similar.

Biochemical data shown suggests GRK2 is a hedgehog regulator, this has been previously published (see my comments below), nevertheless its interesting in functional data by the authors convincingly showing hedgehog pathway dysregulation in patient fibroblasts. Cilia seem ultrastructural intact. So previously published data and results presented here suggest a hedgehog pathway disturbance due to failed smo phosphorylation, the origin is not a ciliary defect but defect of GRK2 essential for Smo phosphorylation. I would therefore omit the term ciliopathy here, this is a "hedgehog-opathy".

Major points:

1. Please change disease label to something like ATD-like ATD-like hedgehog-related chondrodysplasia. I would also suggest to omit the term ciliopathy as there is no primary ciliary defect causing a hedgehog problem but instead a primary smo phosphorylation defect.
2. GRK2 has been suggested to phosphorylate the hedgehog receptor smoothed and hereby act as an essential hedgehog regulator already 15 years ago (Chen et al, Activity-dependent internalization of Smoothed mediated by beta-arrestin 2 and GRK2. Science 306: 2257-2260, 2004). Instead of discussing general functions of G coupled receptors as well as GRK2 functions other than hedgehog signalling in the last part of the introduction, known GRK2 hedgehog pathway functions should be described
3. Figure 4: While a-c convincingly show failed hedgehog signaling pathway activity, panel d is means to show failure of SMO recruitment to the cilium upon stimulation in patient cells compared to control. However the smo staining in general seems a lot weaker in the patient cells, eg background staining nearly absent as well. I am not convinced absence of GRK2 causes failed ciliary smo recruitment based on what I shown in this figure. GRK2 has been previously shown suggested to phosphorylate smo as mentioned above (Science 306: 2257-2260, 2004).
4. GRK-2- mouse model discussed in comparism to the patient: the fact that human survive until birth suggests those are hypomorphs in comparism to the ko mouse. This should be discussed also with regards to the alleles identified

5. Minor points:

Introduction:

1. Page 3 Line 5: While in many publications often a simplified statement that IFT-B governs anterograde IFT and IFT governs retrograde IFT can be found, IFT-A and IFTB form a multiprotein complex involved in both processes in reality. I would suggest to write : " IFT governed by a large multimeric protein complex with 2 main subcomplexes, IFT-A and IFT-B. So-called anterograde IFT

is driven by the kinesin motor KIF3A and mediates transport from the base to the tip of cilia while retrograde IFT is driven by the dynein-2 motor complex and transports cargo from the tip to the base of cilia."

2. Line 12: diverse pleiotropic seems a bit redundant, I would delete diverse here

3. Line 19-23: JATD is probably the mild form of SRPS, being also genetically allelic. I would state this clearly. With regards to multiorgan issues other than skeleton, this is clearly linked to the genotype eg Joubert features with CSPP1, renal/retinal disease with IFT140, IFT144, IFT2, death due to cardiorespiratory issues not occurring with IFT mutations but only dynein- 2 genes

4. Page 4 line 10: "were identified..": probably should read "we identified"

Results:

1. Figure 1: Its is hard see the pelvis in the displayed x-rays, can the authors comment on pelvis configuration, especially if this had JATD typical trident appearance?

2. Figure 2: Its difficult to relate the alleles on protein level (part C) to DNA level (a,b), please indicate which cDNA change corresponds to which protein change either in the legend or in part a/b

3. Figure 3: its unclear to me what "control Ab: non-reactive antibody" means: is this an antibody not directed towards GRK2 or is it supposed to not work in human tissues or does it lack HRP activity? Please specify. In my view, an appropriate control would be secondary antibody without prior primary antibody for example.

Editor comments

Comment: In particular, it will be important to study the defects in Hh and/or Wnt signaling using patient samples (as suggested by reviewer #2), and to provide more mechanistic insights into the failed ciliary SMO localization in human GRK2^{-/-} cells (as commented by reviewer #1). Moreover, attention should be given to placing the study in the context of existing literature.

Answer: All referee comments were addressed in the manuscript revision. We added several additional models to the manuscript, documenting the Hh and Wnt pathway defects in the absence of GRK2. The mechanisms of GRK2 role in Hh and Wnt signaling were also addressed, and the study was placed into the context of existing literature. The authors wish to thank to the editor and referees for their insight and helpful comments that make the article a significantly better story.

Comment: Please make sure that the changes are highlighted to be clearly visible.

Answer: All changes made during the manuscript revision are highlighted in green.

Comment: A .docx formatted letter INCLUDING the reviewers' reports and your detailed point-by-point responses to their comments. As part of the EMBO Press transparent editorial process, the point-by-point response is part of the Review Process File (RPF), which will be published alongside your paper.

Answer: Because this letter contains information about a research we are currently running or are planning to do, we prefer the document not to be public.

Comment: Please note that all corresponding authors are required to supply an ORCID ID for their name upon submission of a revised manuscript.

Answer: The ORCID IDs were added to the corresponding author's accounts.

Comment: We would also encourage you to include the source data for figure panels that show essential data. Numerical data should be provided as individual .xls or .csv files (including a tab describing the data). For blots or microscopy, uncropped images should be submitted (using a zip archive if multiple images need to be supplied for one panel). Additional information on source data and instruction on how to label the files are available at <http://embomolmed.embopress.org/authorguide#sourcedata>.

Answer: We have uploaded two Excel files containing numerical data used to make the graphs, and two PDF documents containing uncropped blots and original microscopy images for all blots and images shown in the main figures and in the Expanded View figures.

Comment: We replaced Supplementary Information with Expanded View (EV) Figures and Tables that are collapsible/expandable online. A maximum of 5 EV Figures can be typeset. EV Figures should be cited as 'Figure EV1, Figure EV2' etc... in the text and their respective legends should be included in the main text after the legends of regular figures.

Answer: We have replaced the Supplementary figures by EV Figures. The revised manuscript contains five EV Figures, annotated in the manuscript text as Figure EV1-5, and one Expanded View Table (Table EV1, originally Supplementary table 1).

Comment: For the figures that you do NOT wish to display as Expanded View figures, they should be bundled together with their legends in a single PDF file called *Appendix*, which should start with a short Table of Content. Appendix figures should be referred to in the main text as: "Appendix Figure S1, Appendix Figure S2" etc.

Answer: There is no Appendix file coupled with the revised version of the manuscript.

Comment: The paper explained: EMBO Molecular Medicine articles are accompanied by a summary of the articles to emphasize the major findings in the paper and their medical implications for the non-specialist reader. Please provide a draft summary of your article highlighting the medical issue you are addressing, the results obtained and their clinical impact.

Answer: We have included The Paper Explained paragraphs in the revised manuscript, right after the Conflict of Interest statement (page 23, paragraphs 3-5).

Comment: Author contributions: the contribution of every author must be detailed in a separate section (before the acknowledgments).

Answer: Author contributions are detailed in the revised manuscript (page 22, paragraph 2).

Comment: A Conflict of Interest statement should be provided in the main text.

Answer: We have provided the Conflict of Interest statement, right after Acknowledgement and Funding Statement (page 23, paragraph 2).

Comment: Every published paper now includes a 'Synopsis' to further enhance discoverability. Synopses are displayed on the journal webpage and are freely accessible to all readers. They include a short stand first (maximum of 300 characters, including space) as well as 2-5 one-sentences bullet points that summarizes the paper. Please write the bullet points to summarize the key NEW findings. They should be designed to be complementary to the abstract - i.e. not repeat the same text. We encourage inclusion of key acronyms and quantitative information (maximum of 30 words / bullet point). Please use the passive voice. Please attach these in a separate file or send them by email, we will incorporate them accordingly. Please also suggest a striking image or visual abstract to illustrate your article. Please provide a jpeg file 550 px-wide x 400-px high.

Answer: We have uploaded a Word document containing a short stand and bullet point summary of the key findings of the paper, as well as a graphical abstract file.

Reviewer's comments

Referee #1 (Remarks for Author):

Question/comment: Surprisingly, the authors make no mention of the prior analysis of GRK2 in Hh signaling at this point, presenting their data as demonstrating "loss of GRK2 compromises HH signaling in cells, rendering them unresponsive to HH pathway activation" as though this were a novel finding (page 8 lines 11/12). This is extremely misleading and unfair to previous authors: the prior studies of GRK2 in Hh signalling should be referred to on page 7 as the basis for investigating disruption of Hh signaling in the patient-derived cells.

Answer: In the revised manuscript, we made sure that the previously published literature on GRK2 function in mammalian SMO and Hh biology has been acknowledged. Additional citations to 9 studies complemented the revised manuscript (mostly found at page 13, paragraph 2). All updated text has been highlighted in green.

Question/comment: The main finding of this study is that loss of GRK2 activity in human cells inhibits the localisation of SMO to the primary cilium (PC) in response to the SMO agonist SAG. This is a surprising finding, given previous findings (confirmed by the authors) that GRK2 activity is not required for Smo localisation in murine cells. This conclusion is supported by experiments in which wild-type chondrocytes treated with GRK2 inhibitors also showed an impairment of SMO localisation to the PC. However, this effect was not absolute:

in this regard, Figure 5a is somewhat misleading as it shows a complete absence of SMO in the PC of treated cells - there is no information about the concentration of inhibitor used on this particular sample - and Figure 5b shows that even at the highest dose, SMO still accumulates in about 35% of cases. It would also be of interest to know if the levels of SMO accumulation are altered in these cells, as appear to be the case for the NIH3T3 cells treated with the same inhibitors (Fig S2 C, E).

Answer: In agreement with the referee comment, the quantifications of SMO signal in the primary cilium were carried out whenever applicable, including all original experiments but also in the new experiments included in the revised manuscript. We think that these quantifications represent the observed phenotypes more faithfully, when compared to original manuscript data showing simple percentages of SMO positive cilia. In the revised manuscript, the quantifications of SMO signal were therefore used to replace the original data (Figs. 4G; 5B, F; EV2B; EV3A, C, E, G, J).

We agree with the referee that, although the SMO quantity is significantly reduced in the primary cilia of R00-082 chondrocytes treated with GRK2 inhibitors, the absence of SMO signal shown by the original Fig. 5A is misleading. In the revised manuscript, new pictures were taken and show a decrease in SMO intensity corresponding to the quantification; the exact concentrations of paroxetine and CMPD101 are also shown in the figure legend. Regarding the NIH3T3 cells treated with GRK2 inhibitors (originally Fig. S2C, E), the quantification of ciliary SMO showed no differences between the SAG-stimulated cells with or without the GRK2 inhibitor; the SAG-naïve cells however show significantly more ciliary SMO in CMPD101-treated cells, which corresponds to the percentages of cilia scored SMO positive in the original manuscript. We have replaced the cilia images by more representative ones; the data are now presented in Figs. 5A; EV2A; EV3C, E.

In the revised manuscript, we used three additional models to address the GRK2 role in SMO translocation to cilia: (1) *Grk2*^{-/-} NIH3T3 cells differentiated to chondrocytes in the micromass culture, and (2) murine IMCD3 cells and (3) human R00-082 chondrocytes stable transfected with doxycycline-inducible shRNA targeting *Grk2* expression. Analyses of these models brought the following findings:

(1) The *Grk2*^{-/-} NIH3T3 micromasses differentiated into chondrocytes *did not* translocate SMO to cilia upon SAG treatment, while the wildtype (*Grk2*^{+/+}) micromasses retained this ability, which is in a striking contrast with the undifferentiated NIH3T3 cells where SMO translocation did not depend on *Grk2* expression.

(2) In murine IMCD3 cells, approximately 41% *Grk2* downregulation by shRNA *did not* produce SAG-induced ciliary SMO signal different from the scramble shRNA controls.

(3) In human R00-082 chondrocytes, the ~74% GRK2 downregulation by shRNA was sufficient to block the SAG-mediated SMO translocation to cilia.

Together, the evidence accumulated in the revised manuscript demonstrates the following: In human cells, the GRK2 loss (in R05-365A patient skin fibroblasts), the shRNA-mediated GRK knock-down (in R00-082 chondrocytes) and inhibition of the GRK2 kinase activity by two kinase inhibitors (in R00-082 and R92-284 chondrocytes) *all inhibited* the SAG-mediated SMO translocation to cilia. In mouse cells, the SMO translocation was abrogated only at one case, i.e. in *Grk2*^{-/-} NIH3T3 cells that were primed towards chondrocytes in micromass culture. In undifferentiated NIH3T3 cells, in NIH3T3 cells treated with *Grk2* inhibitors or in IMCD3 cells after the shRNA-mediated *Grk2* knock-down, the SMO translocated to cilia upon SAG normally. This suggests that the ciliary SMO translocation requires GRK2 in a cell context-dependent manner, and that, in humans, the proper SMO localization to cilia is more sensitive to the GRK2 activity. Notably, the SAG-mediated GLI3 activation and GLI1 induction were inhibited in all mouse and human cell

models used, suggesting the Hedgehog signaling is impaired in the absence of GRK2 or its activity, regardless of SMO translocation.

The NIH3T3 micromass data were added to the revised manuscript as Fig. EV3G, H. The IMCD3 *Grk2* shRNA data were added to the revised manuscript as Fig. EV3I, J. The human R00-082 chondrocytes GRK2 shRNA data were added to the revised manuscript as Fig. 5E, F. Revised manuscript text was updated to accommodate these changes (page 10, paragraphs 1, 2).

Question/comment: In any event, these data in themselves do not support the statement made in the Discussion (Page 10 line 29) "that GRK-phosphorylation of SMO is required to activate the pathway". Such a conclusion requires that the authors investigate the phosphorylation of SMO in their mutant cells, something they do not report.

Answer: In agreement with the referee's comment, the SMO phosphorylation was tested in the R05-365A patient cells and in the *Grk2*^{-/-} NIH3T3 cells. We were not able to detect the endogenous SMO in western blots with three different commercial antibodies (MC2668, MBL International; 20C6, Developmental Studies Hybridoma Bank; E-5, Santa Cruz Biotechnology). We therefore took advantage of the expressed SMO and looked at its phosphorylation status using the phosphoshift PAGE followed by standard western blot. The data demonstrate that loss of GRK2 protein causes inhibition of SMO phosphorylation in both R05-365A patient cells (~35% in SAG-naïve and ~39% in SAG-treated) and the *Grk2*^{-/-} NIH3T3 cells (~46% in SAG-naïve and ~57% in SAG-treated). The SMO phosphorylation data were added to the revised manuscript as Fig. 4D and EV3K; the manuscript text was updated to accommodate these changes (page 9, paragraph 3; page 10, paragraph 2).

In the revised manuscript, the sentence containing "that GRK-phosphorylation of SMO is required to activate the pathway" was removed. The precise mechanism of this process is beyond the scope of the current study, and is opened for future investigation, as discussed in the revised manuscript (page 13).

Question/comment: The authors also present evidence that loss of GRK2 compromises the response of cells to WNT signaling; in particular they describe a reduction in the levels of phosphorylated LRP6 (page 9 line 28); however, this appears to reflect an overall reduction in LRP levels and does not support the statement (page 12 line 25/26) that "GRK2 can also activate Wnt signaling via LRP6 phosphorylation).

Answer: Additional evidence documenting effect of GRK2 inhibition on LRP6 phosphorylation was added to the revised manuscript. We looked at the Wnt3A-mediated LRP6 phosphorylation in rat chondrosarcoma (RCS) chondrocytes and human R00-082 chondrocytes treated with the GRK2 inhibitor CMPD101. CMPD101 abolished Wnt3A-mediated LRP6 phosphorylation without significant alteration of the levels of total LRP6 in both RCS and R00-082 cells. These data were added to revised manuscript as Fig. 7C, D. Revised manuscript text was updated to accommodate these changes (page 11, paragraph 2).

Because the GRK2 loss in the R05-365A patient fibroblasts and *Grk2*^{-/-} NIH3T3 cells manifested as downregulation of the LRP6 protein levels, we aimed to see if the levels of the Frizzled (FZD) Wnt receptor were also affected in these cells. The western blots showed that both cell types express FZD5, but not FZD4 or FZD10, and that the loss of GRK2 does not affect FZD5 expression (revised manuscript Fig. EV5).

We agree with the referee that the statement "GRK2 can also activate Wnt signaling via LRP6 phosphorylation" is misleading, because we did not show that GRK2 is a LRP6 kinase. During the manuscript revision, we carried-out 293T and RCS cell transfections with LRP6 and GRK2, but detected no increased LRP6 phosphorylation in the presence of GRK2 (not shown), suggesting that GRK2 is probably *not* a LRP6 kinase. The statement that "GRK2 can

also activate Wnt signaling via LRP6 phosphorylation” was removed, and the revised manuscript text was updated to indicate that GRK2 regulates Wnt3A by other mechanism than direct LRP6 phosphorylation (page 15, paragraph 1). To interrogate this mechanism, we focused on beta-arrestin (ARRB2), which is well known partner of GRK2, involved in GRK2-mediated agonist-induced desensitization and internalization of b2-adrenergic receptors to endosomes in a clathrin-dependent manner (Goodman et al., 1996). Importantly, ARRB2 also is also a necessary component of the canonical WNT signaling. ARRB2 associates with the transmembrane WNT receptor complex containing LRP6, FZD and disheveled. Because ARRB2 association with WNT signaling complex is necessary for proper activation of WNT signaling in response to WNT ligands (Bryja et al., 2007), we asked whether ARRB2 interaction with FZD is affected by the GRK2 loss. The FZD4 and ARRB2 were expressed in R05-365A patient cells and *Grk2*^{-/-} NIH3T3 cells, and their interaction was probed by proximity ligation assay. The results show loss of interaction of FZD4 with ARRB2 in the GRK2-null cells. This was rescued by *Grk2* addback into the *Grk2*^{-/-} NIH3T3 cells. Our data suggest that one mechanism how GRK2 regulates WNT signaling is promotion of FZD4 interaction with ARRB2. The FZD4/ARRB2 interaction studies were added to the revised manuscript as Figs. 6D; 7E, F. Manuscript text was updated to accommodate new data (page 12, paragraph 2).

Finally, the western blots showing changes in the LRP6 and DVL2 levels and phosphorylation in patient cells (Fig. 6B) were replaced by a new experiment, which improved the visual quality of the presented data; these additional experiments were also included in the densitometry analysis, presented in Fig. 6C.

References

Goodman OB Jr, Krupnick JG, Santini F, Gurevich VV, Penn RB, Gagnon AW, Keen JH, Benovic JL. Beta-arrestin acts as a clathrin adaptor in endocytosis of the beta2-adrenergic receptor. *Nature*. 1996; 383(6599):447-50.

Bryja V, Gradl D, Schambony A, Arenas E, Schulte G. Beta-arrestin is a necessary component of Wnt/beta-catenin signaling in vitro and in vivo. *Proc Natl Acad Sci USA*. 2007; 104(16):6690-6695.

Question/comment: In conclusion, this paper provides data consistent with, but not provide definitive proof of, a causal relationship between loss of GRK2 and ATD. A striking finding is the inhibition of SMO localization to the PC in patient derived cells. Uncovering the basis of this difference would be of great interest but the authors simply comment that its basis "remains unclear" (page 8 line 29). Without further investigation of this difference, it is questionable whether this report is of sufficient interest to merit publication in EMBO Molecular Medicine.

Answer: In the revised manuscript, the relationship between the loss of GRK2 and ATD is supported by the following lines of evidence:

(1) Genetic analyses support that biallelic or homozygous mutations in GRK2 lead to asphyxiating thoracic dystrophy. Loss of function in GRK2 is not tolerated in the human population based on gnomAD data that assigns a loss of function score (LOF) of 1, supporting pathogenesis.

(2) The evidence of impaired Hh signaling in R05-365A patient cells (Fig. 4), in *Grk2*^{-/-} NIH3T3 cells and micromasses (Fig. EV3A-H, K), two human chondrocyte lines R00-082 and R92-284 with GRK2 activity inhibited by two kinase inhibitors (Figs. 5A-D; EV2), and in R00-082 chondrocytes with shRNA-mediated GRK2 downregulation (Fig. 5E, F).

(3) The evidence of deregulation of another pathway important for skeletogenesis, the canonical Wnt signaling, in R05-365A patient cells (Fig. 6), in *Grk2*^{-/-} NIH3T3 cells (Fig. 7A,

B, E, F), and in human R00-082 and R92-284 chondrocytes and RCS cells treated with the GRK2 inhibitors (Figs. 7C, D; EV4).

(4) The evidence of proteoglycan under-sulfation in growth plate cartilage of R05-365A patient (Fig. 3D), chondrogenic micromasses produced from *Grk2*^{-/-} NIH3T3 cells (Fig. 3F), and chicken limb bud micromasses treated with the GRK2 inhibitor (Fig. 3E).

(5) The evidence of *in vivo* shortening of long bones in the chicken wings injected with GRK2 inhibitor (Fig. 3C).

We believe these data identify GRK2 as an important regulator of skeletogenesis. Although the detailed molecular mechanisms of this phenotype are beyond the scope of the current article, we think the topic is worth pursuing further, mainly through analyses of the conditional *Grk2* knockout mice. The work on design of the *Grk2* knockout targeted to bone and cartilage is currently ongoing in the lab.

Question/comment: Another important issue that is not raised is why the loss of SMO activity caused by the loss of GRK2 activity does not have more profound developmental: loss of HH signaling, for instance, is well known to cause major defects in CNS patterning, resulting in holoprosencephaly.

Answer: This is an interesting comment. There were no structural CNS abnormalities noted in the three affected individuals, though two of them succumbed in the postnatal period, thus long-term developmental effects are unknown. However, loss of SMO does indeed cause abnormalities in brain morphogenesis (at 50% reduction, Curry Jones syndrome). In addition, a recent paper (Le et al., 2020) showed several SMO variants that differed in their ability to traffic to the cilium and to activate Hh signaling, producing a wide spectrum of clinical features involving skeleton and CNS.

References

Le T-L, Sribudiani Y, Dong X, et al. Bi-allelic Variations of SMO in Humans Cause a Broad Spectrum of Developmental Anomalies Due to Abnormal Hedgehog Signaling. *Am J Hum Genet.* 2020; 106(6):779-792.

Question/comment: The authors do hint at a possible specific role for GRK2 in skeletogenesis (page 11 line 21) but this is not explored further.

Answer: The evidence presented here demonstrates a lethal skeletal dysplasia caused by loss of GRK2. In the revised manuscript, we have added data on the effect of GRK2 inhibition on chick limb development *in vivo* (Fig. 3C), and the lack of proteoglycan sulfation in the cartilage extracellular matrix in R05-365A patient (Fig. 3D). This observation was confirmed in two independent *in vitro* models, i.e. the *Grk2*^{-/-} NIH3T3 cells differentiated to cartilage micromasses (Fig. 3F), and chicken limb bud micromasses treated with the GRK2 inhibitor (Fig. 3E).

In the revised manuscript, the relationship between the loss of GRK2 and development of lethal skeletal dysplasia is supported by the following lines of evidence:

(1) Genetic analyses support that biallelic or homozygous mutations in GRK2 lead to asphyxiating thoracic dystrophy. Loss of function in GRK2 is not tolerated in the human population based on gnomAD data that assigns a loss of function score (LOF) of 1, supporting pathogenesis.

(2) The evidence of impaired Hh signaling in R05-365A patient cells (Fig. 4), in *Grk2*^{-/-} NIH3T3 cells and micromasses (Fig. EV3A-H, K), two human chondrocyte lines R00-082 and R92-284 with GRK2 activity inhibited by two kinase inhibitors (Figs. 5A-D; EV2), and in R00-082 chondrocytes with shRNA-mediated GRK2 downregulation (Fig. 5E, F).

(3) The evidence of deregulation of another pathway important for skeletogenesis, the

canonical Wnt signaling, in R05-365A patient cells (Fig. 6), in *Grk2*^{-/-} NIH3T3 cells (Fig. 7A, B, E, F), and in human R00-082 and R92-284 chondrocytes and RCS cells treated with the GRK2 inhibitors (Figs. 7C, D; EV4).

(4) The evidence of proteoglycan under-sulfation in growth plate cartilage of R05-365A patient (Fig. 3D), chondrogenic micromasses produced from *Grk2*^{-/-} NIH3T3 cells (Fig. 3F), and chicken limb bud micromasses treated with the GRK2 inhibitor (Fig. 3E).

(5) The evidence of *in vivo* shortening of long bones in the chicken wings injected with GRK2 inhibitor (Fig. 3C).

Our data identify GRK2 an important regulator of skeletal tissues, particularly the proteoglycan synthesis in chondrocytes. The mechanism of this phenotype is beyond the scope of the current article, but we think it is worth pursuing further, mainly through analyses of the conditional *Grk2* knockout mice. The work on design of the *Grk2* knockout in bone and cartilage is currently ongoing in the lab.

Referee #2

Comments on Novelty/Model System

The major limitation of this study is the absence of an animal model to support the loss of GRK2 as the cause of skeletal phenotype in the patients. This is because complete loss of GRK2 is embryonic lethal in both mice and zebrafish. Theoretically, investigators could try to accomplish a conditional mouse model (with expression limited to cartilage) or studying cartilage development in zebrafish embryos, etc. However, to be practical, these models may not necessarily be adequate (for example a conditional mouse model could be early lethal as well) and I do appreciate the strong *in vitro* data that the authors included to support their conclusion.

Answer: Despite being embryonically lethal, the *Grk2* knockout mice were reported to live up to E15.5 (Jaber et al., 1996). This is after the onset of skeletogenesis, which starts at E11.5 for cartilage formation and E15.5 for ossification, respectively. We thus thought we could use the *Grk2*^{-/-} mice to probe the role of Grk2 in early stages of cartilage/bone development. We obtained the previously published *Grk2*^{-/-} mice (Jaber et al., 1996) in 2015, and maintained the stock for 3 years. During this time, we tried to recover *Grk2*^{-/-} embryos multiple times but were never able to obtain embryos older than E9.5, i.e. before the onset of cartilage formation. We also failed to establish the *Grk2*^{-/-} MEF lines. Although the *Grk2*^{-/-} MEFs could be isolated, they grew past the primary culture, thus being useless for experimentation.

In the revised manuscript, we report on the lack of proteoglycan sulfation in the cartilage extracellular matrix in R05-365A patient (Fig. 3D). This observation was confirmed in two independent *in vitro* models, i.e. the *Grk2*^{-/-} NIH3T3 cells differentiated to cartilage micromasses (Fig. 3F), and chicken limb bud micromasses treated with the GRK2 inhibitor (Fig. 3E). Our data suggest that GRK2 is a regulator of the proteoglycan synthesis in chondrocytes. The mechanism of this phenotype is beyond the scope of the current article, but we think it is worth pursuing further, mainly through analyses of the conditional *Grk2* knockout mice. The work on design of the *Grk2* knockout in bone and cartilage is currently ongoing in the lab.

References

Jaber M, Koch WJ, Rockman H, Smith B, Bond RA, Sulik KK, Ross J Jr, Lefkowitz RJ, Caron MG, Giros B, Essential Role of Beta-Adrenergic Receptor Kinase 1 in Cardiac Development and Function. *Proc Natl Acad Sci USA*.1996;93(23):12974-9.

Remarks for Author

The paper reports two families with asphyxiating thoracic dystrophy (ATD) and pathogenic variants in GRK2. A thorough clinical description, including radiographs and a detailed clinical table, supports the patients' phenotype to be consistent with a skeletal ciliopathy, specifically ATD. The investigators performed *in vitro* mechanistic studies using patient fibroblasts and cell lines to show that disruption of Hedgehog signaling (via impaired translocation of smoothened to the cilia) and abnormal canonical Wnt signaling (due to decreased levels and reduced phosphorylation of LRP6) contribute to the phenotype in GRK2-null cells. The manuscript contains novel data that is clinically relevant and of broad significance by describing a new form of skeletal dysplasia that is mediated by loss-of-function of GRK2, a ser/thr kinase. There are however a few issues that should be addressed by the authors.

Major Comments

Question/comment: 1. The major weakness of this study is the lack of *in vivo* data to further support that loss of GRK2 is causing the skeletal phenotype in the patients. I realize this is difficult to overcome since GRK2^{-/-} mice and zebrafish show early embryonic lethality (necessitating a conditional model or studies in early embryonic stage). The authors may consider citing in the discussion previous paper showing severe growth retardation and abnormal limb development in the knockout mouse model (PMID 18815277).

Answer: We appreciate the referee's suggestion and have included the information on the growth retardation observed in the *Grk2*^{-/-} mice (PMID18815277; Philipp et al., 2008) into the revised manuscript (page 6, paragraph 2). As described in detail above, we tried to use the general *Grk2* knockout mice to address the role of GRK2 during the early stages of skeletal development. This was not successful because the *Grk2*^{-/-} embryos died before the onset of skeletogenesis. We are currently designing the conditional cartilage *Grk2* knockout model, however its analyses are beyond the scope of the current article.

The revised manuscript was strengthened by adding the following new data:

1. SMO translocation to primary cilia, in three different cell models (Figs. 5E, F; EV3G-J).
2. SMO phosphorylation in two GRK2-deficient cell models (Figs. 4D; EV3K).
3. LRP6 phosphorylation in two cell lines treated with GRK2 inhibitor (Fig. 7C, D).
4. FZD4 interaction with ARRB2 in GRK2 deficient cells (Figs. 6D; 7E, F).
5. Evidence of chondrocyte extracellular matrix under-sulfation in patient cells and two independent *in vitro* models (Fig. 3D-F).
6. Evidence on the effect of GRK2 inhibition on chick limb growth *in vivo* (Fig. 3C).

References

Philipp M, Fralish GB, Meloni AR, et al. Smoothened signaling in vertebrates is facilitated by a G protein-coupled receptor kinase. *Mol Biol Cell*. 2008; 19(12):5478-5489.

Question/comment: 2. As an alternative to animal model, and since cartilage growth plate is available from one of the subjects, the authors may consider studying the defect in Hh and/or wnt signaling in the affected tissue (for example by immunohistochemistry for components of the signaling pathway or known downstream targets).

Answer: The immunohistochemistry on several targets of WNT and Hh signaling was tried but failed, possibly due to the long storage (more than 10 years) and overfixation of the sample. The growth plate cartilage samples available on the R05-365A patient were stained with alcian blue, to visualize the sulfated proteoglycan content of the chondrocyte extracellular matrix. At pH2.5 (proteoglycan staining) (Green and Pastewka, 1974), we found no significant changes in the signal between control and R05-365A growth plates. At pH1.0

(proteoglycan staining), a lack of staining was found in R05-365A, suggesting proteoglycan under-sulfation (Fig. 3D).

The putative lack of proteoglycan sulfation in the absence of GRK2 or its kinase activity, was confirmed in two independent *in vitro* models, i.e. the *Grk2*^{-/-} NIH3T3 cells differentiated to cartilage (Fig. 3F), and chick limb bud micromass cultures treated with GRK2 inhibitor (Fig. 3E). Our data suggest previously unappreciated role of GRK2 in regulation of growth plate cartilage. The mechanism of this phenotype is beyond the scope of the current article, and will be addressed in the follow up study.

Reference

Green MR, Pastewka JV. Simultaneous differential staining by a cationic carbocyanine dye of nucleic acids, proteins and conjugated proteins. II. Carbohydrate and sulfated carbohydrate-containing proteins. *J Histochem Cytochem.* 1974; 22(8):774-81.

Minor comments

Question/comment: The detailed clinical description is much appreciated. I suggest to edit and make the styling more consistent between the paragraphs of two families. Also, authors may consider replacing "case" (in abstract) with "proband" or "affected individual" similar to what they used later in the text.

Answer: Thank you. We replaced the “case” with “proband” or “affected individual” through the manuscript text.

Question/comment: It will be helpful to uniformly describe the variants using HGVS nomenclature, including hg19 genomic position, and authors may consider replacing the term "mutation" with "variant" or "pathogenic variant".

Answer: Thank you. We have changed the term mutation to “variant” or “pathogenic variants” in portions of the manuscript where it was appropriate. The term “mutations” remain in a few spots based on the authors believing that it was most term.

Referee #3

Comment: This is well written manuscript describing a novel, ATD-like human phenotype due to biallelic GRK2 loss of function mutations. This will be of interest to a broad community of developmental biologists as well as human geneticists. The genetic data is convincing and functional data also of good quality. The authors suggest the phenotype observed is JATD and that GRK2 loss of function causes a ciliopathy. The genetic data is convincing that the patient phenotype is a result of GRK2 loss of function. The phenotype resembles ATD/skeletal ciliopathies with regards to short ribs and long bones, however doesn't seem to be identical (eg no typical pelvis configuration, no polydactyly). Short ribs occur in different human disease and are not limited to ciliopathies eg can be observed in thanophoric dysplasia (FGFR3 variants) or Schachman-Diamond S. (SBDS variants). I would therefore suggest to label the phenotype an ATD-like hedgehog-related chondrodysplasia or similar.

Answer: Thank you, throughout the paper we have used the term ATD.

Comment: Biochemical data shown suggests GRK2 is a hedgehog regulator, this has been previously published (see my comments below), nevertheless its interesting in functional data by the authors convincingly showing hedgehog pathway dysregulation in patient fibroblasts. Cilia seem ultrastructural intact. So previously published data and results presented here suggest a hedgehog pathway disturbance due to failed smo phosphoryation, the origin is not a

ciliary defect but defect of GRK2 essential for Smo phosphorylation. I would therefore omit the term ciliopathy here, this is a "hedgehog-opathy".

Answer: We appreciate the suggestion and think it is thoughtful. However, in addition to hedgehog signaling, WNT signaling and sulfation of cartilage proteoglycans are affected so we think that referring to this phenotype as a "hedgehog-opathy" is too narrow.

Comment/question: Major points:1. Please change disease label to something like ATD-like ATD-like hedgehog-related chondrodysplasia. I would also suggest to omit the term ciliopathy as there is no primary ciliary defect causing a hedgehog problem but instead a primary smo phosphorylation defect.

Answer: Thank you. We agree that our data showed that ciliogenesis and cilia architecture are not affected due to alterations in GRK2. In all areas of the text we changed the phenotype to ATD where appropriate. There are a few general references to ATD as skeletal ciliopathy. We think it is important that downstream the phenotype be recognized as ATD because clinicians need to include GRK2 in the genes that produce the ATD spectrum of disease.

Comment/question: 2. GRK2 has been suggested to phosphorylate the hedgehog receptor smoothed and hereby act as an essential hedgehog regulator already 15 years ago (Chen et al, Activity-dependent internalization of Smoothed mediated by beta-arrestin 2 and GRK2. Science 306: 2257-2260, 2004). Instead of discussing general functions of G coupled receptors as well as GRK2 functions other than hedgehog signalling in the last part of the introduction, known GRK2 hedgehog pathway functions should be described.

Answer: In agreement with referee's comment, we have included paragraph detailing the published data on GRK2 function in regulation of the Hh pathway, and how it relates to the present study (page 13, paragraphs 1, 2).

Comment/question: 3. Figure 4: While a-c convincingly show failed hedgehog signaling pathway activity, panel d is means to show failure of SMO recruitment to the cilium upon stimulation in patient cells compared to control. However the smo staining in general seems a lot weaker in the patient cells, eg background staining nearly absent as well. I am not convinced absence of GRK2 causes failed ciliary smo recruitment based on what I shown in this figure. GRK2 has been previously shown suggested to phosphorylate smo as mentioned above (Science 306: 2257-2260, 2004).

Answer: In agreement with the referee comment, we carried-out quantifications of SMO signal in the primary cilium whenever applicable, including original experiments and new experiments produced during the revision (Figs. 4G; 5B, F; EV3A, C, E, G, J; EV2B). These quantifications of SMO signal in cilia replaced the original data showing simply percentages of SMO positive cilia. New representative images were used when necessary (Figs. 4E; 5A; EV3C, E; EV2A), to better represent the now quantitative measurement of SMO presence in cilia.

In the revised manuscript, we used three additional models to address the GRK2 role in SMO translocation to cilia: (1) *Grk2*^{-/-} NIH3T3 cells differentiated to chondrocytes in the micromass culture, and (2) murine IMCD3 cells and (3) human R00-082 chondrocytes stable transfected with doxycycline-inducible shRNA targeting *Grk2* expression. Analyses of these models brought the following findings:

(1) The *Grk2*^{-/-} NIH3T3 micromasses differentiated into chondrocytes *did not* translocate SMO to cilia upon SAG treatment, while the *Grk2*^{+/+} micromasses retained this ability, which is in a striking contrast with the undifferentiated NIH3T3 cells where SMO translocation did not depend on *Grk2* expression.

(2) In murine IMCD3 cells, approximately 41% Grk2 downregulation by shRNA *did not* produce SAG-induced ciliary SMO signal different from the scramble shRNA controls.

(3) In human R00-082 chondrocytes, the 74% GRK2 downregulation by shRNA was sufficient to block the SAG-mediated SMO translocation to cilia.

We looked at the SMO phosphorylation status using the phosphoshift PAGE followed by standard western blotting. The data demonstrate that loss of the GRK2 causes a similar inhibition of SMO phosphorylation in both human R05-365A patient cells and in the mouse *Grk2*^{-/-} NIH3T3 cells.

Together, the evidence accumulated in the revised manuscript demonstrates the following: In human cells, the GRK2 loss (in R05-365A patient skin fibroblasts), the shRNA-mediated GRK knock-down (in R00-082 chondrocytes) and inhibition of the GRK2 kinase activity by two kinase inhibitors (in R00-082 and R92-284 chondrocytes) *all inhibited* the SAG-mediated SMO translocation to cilia. In mouse cells, the SMO translocation was abrogated only at one case, i.e. in *Grk2*^{-/-} NIH3T3 cells that were primed towards chondrocyte differentiation in micromass culture. In undifferentiated NIH3T3 cells, in NIH3T3 cells treated with Grk2 inhibitors or in IMCD3 cells after the shRNA-mediated Grk2 knock-down, the SMO translocated to cilia upon SAG normally. The SMO phosphorylation is GRK2-dependent, and this phosphorylation was similarly impaired in the GRK2-null R05-365A patient cells and the *Grk2*^{-/-} NIH3T3 cells. Notably, the SAG-mediated GLI3 activation and GLI1 induction were inhibited in all mouse and human cell models used, suggesting the Hedgehog signaling is impaired in the absence of GRK2 activity, regardless of SMO translocation, which was preserved in some mouse models. The proper SMO localization to cilia is therefore more sensitive to the GRK2 activity in humans. In mouse cells lacking GRK2, the SMO may translocate to cilia but cannot activate Hh pathway, suggesting that modifications by GRK2, such as phosphorylation, are essential for Hh signaling.

The NIH3T3 micromass data were added to the revised manuscript as Fig. EV3G, H. The IMCD3 Grk2 shRNA data were added to the revised manuscript as Fig. EV3I, J. The human R00-082 chondrocytes GRK2 shRNA data were added to the revised manuscript as Figure 5E, F. The SMO phosphorylation data were added to the revised manuscript as Figures 4D and EV3K. Revised manuscript text was updated to accommodate these changes (page 9, paragraph 3; page 10, paragraphs 1, 2).

Comment/question: 4. GRK-2- mouse model discussed in comparison to the patient: the fact that human survive until birth suggests those are hypomorphs in comparison to the ko mouse. This should be discussed also with regards to the alleles identified.

Answer: We agree that this is a very interesting point. While we agree this suggests that the humans may be hypomorphic for GRK2 expression, the data derived from the affected patient did not support that at the RNA or protein levels. While we do not have an explanation for the difference lethality in mouse versus human, this has been seen for other mouse models such as FKBP10 which loss causes lethality by E15.5, but affected humans with loss of FKBP10 have osteogenesis imperfecta.

Comment/question: 5. Minor points: Introduction: 1. Page 3 Line 5: While in many publications often a simplified statement that IFT-B governs anterograde IFT and IFT governs retrograde IFT can be found, IFT-A and IFTB form a multiprotein complex involved in both processes in reality. I would suggest to write : " IFT governed by a large multimeric protein complex with 2 main subcomplexes, IFT-A and IFT-B. So-called anterograde IFT is driven by the kinesin motor KIF3A and mediates transport from the base to the tip of cilia while retrograde IFT is driven by the dynein-2 motor complex and transports cargo from the tip to the base of cilia."

Answer: In agreement with the referee's suggestion, we have updated the corresponding paragraph (page 3, paragraph 1).

Comment/question: 2. Line 12: diverse pleiotropic seems a bit redundant, I would delete diverse here.

Answer: The revised manuscript text was updated to accommodate the requested change (page 3, paragraph 2).

Comment/question: 3. Line 19-23: JATD is probably the mild form of SRPS, being also genetically allelic. I would state this clearly. With regards to multiorgan issues other than skeleton, this is clearly linked to the genotype eg Joubert features with CSPP1, renal/retinal disease with IFT140, IFT144, IFT2, death due to cardiorespiratory issues not occurring with IFT mutations but only dynein- 2 genes.

Answer: Thank you. We added the term "phenotypic" to line 24 which hopefully clarifies that many genes produce the spectrum of disease.

Comment/question: 4. Page 4 line 10: "were identified..": probably should read "we identified".

Answer: Thank you, the wording was corrected.

Comment/question: Results: 1. Figure 1: Its is hard see the pelvis in the displayed x-rays, can the authors comment on pelvis configuration, especially if this had JATD typical trident appearance?

Answer: We thank the reviewer for this salient comment. As noted in the text, while trident pelvis is often seen in JATD, it was not seen in our affected individuals and it is not listed in the table. We used the term small pelvis.

Comment/question: 2. Figure 2: Its difficult to relate the alleles on protein level (part C) to DNA level (a,b), please indicate which cDNA change corresponds to which protein change either in the legend or in part a/b

Answer: In agreement with the referee's comment, we have updated the corresponding panels in Figure 2.

Comment/question: 3. Figure 3: its unclear to me what "control Ab: non-reactive antibody" means: is this an antibody not directed towards GRK2 or is it supposed to not work in human tissues or does it lack HRP activity? Please specify. In my view, an appropriate control would be secondary antibody without prior primary antibody for example.

Answer: Instead of simple staining without primary antibody, we perform a non-specific primary antibody control. For the 'Ab control' in original manuscript Fig. 3A, the Purified Rabbit IgG Isotype Standard (BD Biosciences. Ref: 550875) was used instead of the GRK2 primary antibody. The revised manuscript Material and Methods (page 21, paragraph 1) section was updated to prevent confusion.

Additional changes to the manuscript

The manuscript revision resulted in additional data, which was either used to complement the originally submitted figures, or to produce new figures, as detailed above. The following additional changes made to the revised manuscript were made:

(1) Some data originally presented in main figures were moved to the EV Figures as required by the Journal format (specifically, Fig. 7C-F are now Fig. EV4A-D). Most of the

originally Supplementary data has become part of the EV Figures (specifically, Figs. S1 and S3 are now Fig. EV1; Fig. S2 is now Fig. EV3A-F; Fig. S5 is now Figure EV2; Fig. S6A is now a part of Fig. EV4).

(2) The positioning of the growth plate zones in the R05-365A patient was corrected, reflecting their actual distribution (Fig. 3B).

(3) Because the ciliary translocation of SMO, originally presented as the percentage of SMO-positive cells, was replaced by measurements of the ciliary SMO intensity, we also introduced small changes into the figures analyzing the effect of CMPD101 and paroxetine GRK2 inhibitors on ciliary SMO translocation in the human chondrocyte lines R00-082 and R92-284. The previously presented data of a gradually increasing effect of GRK2 inhibitors CMPD101 and paroxetine on percentage of SMO-positive cells and ciliation frequency are now only shown for the highest concentration of the two inhibitors, as they delivered the maximal effect and were also used in the subsequent analyses of the Hh signaling (GLI blots). That resulted in the absence of the data originally presented as the Supplementary figure 4, showing percentage of ciliated cells in CMPD101/paroxetine concentrations not anymore used in the revised manuscript. The updated data are now presented in Figs. 5B, EV2B and EV1G.

(4) We removed data showing the effect of paroxetine on Wnt signaling in two additional chondrocyte lines, R02-137 and R93-064B, because of the low number of replicates that did not fit the Journal requirements.

(5) Additional Gli3 and Gli1 western blots in NIH3T3 cells were produced during the manuscript revision, and the obtained data were used to update the Figure EV3B, D, F.

(6) The list of authors was also updated to accommodate their contribution to the whole manuscript work, including the revisions.

(7) Most of the additional data, produced during the manuscript revision, was based on the comments raised during the original review. In addition, novel data were added to the revised manuscript, showing under-sulfation of the R05-365A patient growth plate cartilage. This observation was replicated in two additional *in vitro* models, i.e. the chondrogenic micromasses produced from *Grk2*^{-/-} NIH3T3 cells, and the chicken limb bud micromasses treated with the GRK2 inhibitor.

Thank you for the submission of your revised manuscript to EMBO Molecular Medicine. We have now received the enclosed report from the two referees who were asked to re-assess it. As you will see below both referees still raise a couple of concerns on your work, which need to be addressed in a revision of the present manuscript. In particular, attention should be given to placing the study in the context of existing literature. Further, referee #1 is still concerned about the link between GRK2, Hh function and skeletal development, and suggested experiments in this regard. We would encourage you to address this concern if the proposed experiments by this referee are feasible, otherwise the limitations should be discussed.

***** Reviewer's comments *****

Referee #1 (Remarks for Author):

In my original review of the report by Bosakova et al on the association between asphyxiating thoracic dystrophy (ATD) or Jeune 18 syndrome, and mutations in the GRK2 gene I expressed a number of reservations about the data and their interpretation:

1. The authors claim that loss of GRK2 activity in human cells inhibits the localisation of SMO to the primary cilium (PC) in response to the SMO agonist SAG is not fully substantiated by the data.

The authors have addressed this point to some extent by providing more quantitative data and performing analyses of three additional cell models. Whilst this enhances their story, the data still do not support the conclusion (p16 line 19) that loss of GRK2 activity results in a constitutive inhibition of the Hh pathway via defective SMO translocation to cilia. As the authors themselves show, loss of GRK2 activity in 3T3 cells can inhibit Hh pathway activity without affecting trafficking of SMO to the cilia and there is nothing to show that the failure of SMO to localize to the cilia in human GRK2 null cells is the cause of Hh pathway inhibition.

2. the statement that "GRK-phosphorylation of SMO is required to activate the pathway" is not supported by analysis of the phosphorylation of SMO in GRK2 mutant cells.

The authors now present evidence that SMO phosphorylation is inhibited in human patient and Grk2 null 3T3 cells. However, they also acknowledge that these data do not establish causality and have removed this claim

3. the claimed effect of loss of GRK2 activity on LRP6 phosphorylation is not well supported and does not substantiate the claim that "GRK2 can also activate Wnt signaling via LRP6 phosphorylation".

The authors have now added data that argue against a direct role of GRK2 in LRP6 phosphorylation and have modified their interpretation accordingly. They had also performed additional experiments that support a role for GRK2 in promoting interaction of FZD4 with ARRB2

4. the report provides data consistent with, but not definitive proof of, a causal relationship between loss of GRK2 and ATD.

The point here is that discovery of mutations in two ATD patients, whilst providing strong evidence for causality, cannot in itself definitively establish this link. This limitation was also raised by Reviewer 2 who highlighted the lack of an animal model to support the loss of GRK2 as the cause of skeletal phenotype in the patients.

To address this issue, the authors have performed some additional analyses:

First, they present evidence of under-sulphation of proteoglycans in the patient growth plate cartilage as well as in chondrogenic micromasses produced from Grk2^{-/-} 3T3 cells as well as chicken limb bud micromasses treated with the GRK2 inhibitor

Second, by injecting GRK2 inhibitor into the limb buds of chick embryos, they have obtained some evidence of a role for GRK2 in promoting the growth of long bones in chicken wings. While these data strengthen the case, more definitive evidence awaits the mouse conditional knock out experiments alluded to by the authors.

One additional line of evidence that might strengthen the link between GRK2, Hh function and skeletal development would be an analysis of PTHrP expression in the patient growth plate. These samples should be susceptible to in situ hybridization analysis - however, I am not an expert in this area and am uncertain if the samples are at an appropriate developmental stage to perform such an analysis.

While these revisions have improved the manuscript, I still have two major reservations.

The first of these concerns the lack of recognition of prior analyses of the role of Grk2 in Hh signalling. As noted also by Reviewer 3, the authors fail to present their analysis in the light of earlier studies, which I find to be bordering on intellectually dishonesty. The obvious place to cite prior analyses would be in the Introduction (as also suggested by Reviewer 3), yet despite pointing out (page 4. line 21) that "GRK2 participates in many other processes" besides desensitizing GPCRs, the authors studiously avoid any reference to its role in Hh signalling. This is a particularly egregious omission, given that they subsequently present data that effectively recapitulate the results of earlier studies: viz. the failure to block processing of the Gli transcription factor to its repressor form and the concomitant loss of target gene (Gli1 and Ptch) in response to Shh or SAG mediated Smo activation in the absence of GK2 activity. All of these effects were shown to result from loss of Grk2 activity in zebrafish embryos and NIH 3T3 cells by Zhao et al (EMBO Reports, 2016), a paper of which two of the authors of the current report are co-authors (and of which I am the senior author)! The authors say they have addressed this issue by adding all relevant references - but they do so in a very elliptical manner and only in the Discussion (page 13). There is certainly no explicit

reference to the zebrafish studies, which constitute the most comprehensive in vivo analysis of the role of GRK2 in Hh signalling to date. Of course, the authors are careful to refer to "mammalian" Hh signalling throughout their report - but they should remember that Hh signalling was first elucidated in *Drosophila*, that Shh was first discovered in zebrafish and that chickens - which they use to validate the effect of GRK2 inhibition in vivo - are not mammals!!

My second reservation concerns the authors persistent failure to address the paradoxical finding that, despite the apparent loss of SMO activity in GRK2 mutant cells, the patient from whom these cells are derived developed to term without manifesting any of the gross defects characteristic of loss of SMO (or SHH). While the authors acknowledge that "This is an interesting comment" and affirm that "there were no structural CNS abnormalities noted in the three affected individuals", they fail to confront this issue in their discussion. As I pointed out the authors do hint at the possibility of a specific role for GRK2 in skeletogenesis, but this is not explored further. By this I meant that they did not consider why loss of GRK2 might specifically affect chondrocytes whilst sparing neural progenitors (and likely other cells). One fact that the authors overlook is that, unlike the zebrafish, mammals possess two apparently functionally redundant GRK2 paralogues, which explains why the mouse *grk2* null phenotype is less extreme than that of the zebrafish. The authors might wish to consider the possibility that their GRK2 mutant chondrocyte lines are compromised in Hh signalling because they do not express the paralogous (GRK3) gene.

Referee #2 (Comments on Novelty/Model System for Author):

The existing animal models are early embryonic lethal and do not allow downstream analyses. The authors are working to develop conditional mouse model but this will be beyond the scope of this study. Given these limitations, the authors provide abundant in vitro data that is well-controlled and recapitulated in several different cell culture models derived from mouse or human samples, using different approaches including knock-down experiments and chemical inhibition, and supported by in vivo/ex vivo chick limb bud model. This reviewer finds the above model systems and experimental approaches to be adequate and sufficient to support the authors hypothesis.

Referee #2 (Remarks for Author):

Summary

The authors submitted a revised manuscript reporting pathogenic variants in GRK2 as the cause of new form of asphyxiating thoracic dystrophy (ATD), a genetically and clinically heterogeneous disorder. In this revised version the authors provided abundant experimental evidence to support the role of GRK2 in skeletogenesis, and to demonstrate that loss of GRK2 is associated with ciliary dysfunction and dysregulated Hh and Wnt signaling pathways contributing to the patients' phenotype. The manuscript is well-written, and contains novel data that is clinically relevant and of broad significance. The authors appropriately addressed this reviewer's concerns, and I only have a few minor suggestions that may make the manuscript better.

Minor comments

1. The authors should consider revising Figure 3 (such that the "flow" of data is consistent with the A to F organization). The abundant data is appreciated but difficult to follow in the current format.
2. In Figure 5B and EV2B the quantification of SMO signal still suggests some translocation of SMO to the cilia even in the presence of inhibitors (that is statistically significant when compared to cells without SAG treatment). I suggest to change the title to express that inhibition of GRK2 activity

partially inhibited, or decreased, rather than completely blocked Smoothed accumulation in the cilia. This will also be consistent with Hh impaired signaling not being the only player but rather one of the causes of abnormal skeletal development, along with dysregulated Wnt signaling (and possibly other pathways as mentioned in the discussion).

3. In Figure EV1A please clarify in the bottom panel whether red staining marks pericentrin or acetylated tubulin.

4. In Figure EV4 while panels C, D and E appear to be similarly treated the graphic presentation is not consistent across different models (specifically the +/- paroxetine and Wnt3A treatments are organized differently in the RCS panel as compared to human chondrocyte cells). This is somewhat confusing and I suggest to revise panel C to match with panels D, E below.

Reviewer's comments

Referee #1 (Remarks for Author) In my original review of the report by Bosakova et al on the association between asphyxiating thoracic dystrophy (ATD) or Jeune 18 syndrome, and mutations in the GRK2 gene I expressed a number of reservations about the data and their interpretation:

Comment: 1. The authors claim that loss of GRK2 activity in human cells inhibits the localization of SMO to the primary cilium (PC) in response to the SMO agonist SAG is not fully substantiated by the data.

The authors have addressed this point to some extent by providing more quantitative data and performing analyses of three additional cell models. Whilst this enhances their story, the data still do not support the conclusion (p16 line 19) that loss of GRK2 activity results in a constitutive inhibition of the Hh pathway via defective SMO translocation to cilia. As the authors themselves show, loss of GRK2 activity in 3T3 cells can inhibit Hh pathway activity without affecting trafficking of SMO to the cilia and there is nothing to show that the failure of SMO to localize to the cilia in human GRK2 null cells is the cause of Hh pathway inhibition.

Answer: It is true that although the inhibition of Hh signaling was observed in all models with impaired GRK2 signaling used in the paper, the defective SMO translocation accompanied loss of GRK2 in five out of eight models. We thank the referee for pointing this out. The manuscript was corrected where a direct causality between the impaired ciliary SMO trafficking and inhibited Hh readouts was suggested (page 18, paragraph 2).

Comment: 2. the statement that "GRK-phosphorylation of SMO is required to activate the pathway" is not supported by analysis of the phosphorylation of SMO in GRK2 mutant cells.

The authors now present evidence that SMO phosphorylation is inhibited in human patient and Grk2 null 3T3 cells. However, they also acknowledge that these data do not establish causality and have removed this claim.

Comment: 3. the claimed effect of loss of GRK2 activity on LRP6 phosphorylation is not well supported and does not substantiate the claim that "GRK2 can also activate Wnt signaling via LRP6 phosphorylation".

The authors have now added data that argue against a direct role of GRK2 in LRP6 phosphorylation and have modified their interpretation accordingly. They had also performed additional experiments that support a role for GRK2 in promoting interaction of FZD4 with ARRB2

Comment: 4. the report provides data consistent with, but not definitive proof of, a causal relationship between loss of GRK2 and ATD. The point here is that discovery of mutations in two ATD patients, whilst providing strong evidence for causality, cannot in itself definitively establish this link. This limitation was also raised by Reviewer 2 who highlighted the lack of an animal model to support the loss of GRK2 as the cause of skeletal phenotype in the patients.

To address this issue, the authors have performed some additional analyses: First, they present evidence of under sulphation of proteoglycans in the patient growth plate

cartilage as well as in chondrogenic micromasses produced from *Grk2*^{-/-} 3T3 cells as well as chicken limb bud micromasses treated with the GRK2 inhibitor. Second, by injecting GRK2 inhibitor into the limb buds of chick embryos, they have obtained some evidence of a role for GRK2 in promoting the growth of long bones in chicken wings. While these data strengthen the case, more definitive evidence awaits the mouse conditional knock out experiments alluded to by the authors. \

Answer: As described in detail in the first revision, we tried to address the skeletal phenotype of *Grk2*^{-/-} mice using a general knockout, but failed to do so over 3 years of trials. We are therefore designing a conditional knockout model, to be used in the follow up study addressing the effect of *Grk2* loss on chondrocyte differentiation and extracellular matrix homeostasis.

In addition, while we respect the reviewer's opinion on causation and limited animal models, the three variants or mutations in the paper have not been seen in the gnomAD database supporting that they are not common alleles in the population. Further all three mutations are predicted to be stop codons, supported by lack of protein shown in one of the cases. The gnomAD database has assigned a pLOF or pLi score of 0.18 which is very low and supports that loss of function mutations are not tolerated and there are no homozygotes with loss of function in the control database.

Comment: One additional line of evidence that might strengthen the link between GRK2, Hh function and skeletal development would be an analysis of PTHrP expression in the patient growth plate. These samples should be susceptible to in situ hybridization analysis - however, I am not an expert in this area and am uncertain if the samples are at an appropriate developmental stage to perform such an analysis.

Answer: We agree with the referee that analyses of expression of components of Hh, Wnt and PTHrP signaling pathways in the patient growth plate could bring supporting evidence linking the molecular mechanisms to the tissues phenotypes. That was however not possible due to the condition of the ATD patient growth plate, which was over 10 years in the storage and possibly overfixed. As the general *Grk2*^{-/-} knockout mice died before the onset of cartilage in our hands, we could not perform these analyses using fresh mouse tissues. The expression of the components of Hh, Wnt and PTHrP pathways, and the targets of their signaling, will be addressed by *in situ* hybridization and immunohistochemistry in the conditional *Grk2* knockout model we are currently designing, and hopefully reported in the follow up article.

Comment: While these revisions have improved the manuscript, I still have two major reservations. The first of these concerns the lack of recognition of prior analyses of the role of *Grk2* in Hh signalling. As noted also by Reviewer 3, the authors fail to present their analysis in the light of earlier studies, which I find to be bordering on intellectually dishonesty. The obvious place to cite prior analyses would be in the Introduction (as also suggested by Reviewer 3), yet despite pointing out (page 4. line 21) that "GRK2 participates in many other processes" besides desensitizing GPCRs, the authors studiously avoid any reference to its role in Hh signalling. This is a particularly egregious omission, given that they subsequently present data that effectively recapitulate the results of earlier studies: viz. the failure to block processing of the GLI transcription factor to its repressor form and the concomitant loss of target gene (*Gli1* and *Ptch*) in response to *Shh* or *SAG* mediated *Smo* activation in the absence of *GK2* activity. All of these effects were shown to result from loss of *Grk2* activity in zebrafish embryos and NIH 3T3 cells by Zhao et al (EMBO Reports, 2016), a paper of which two of the authors of the current report are co-authors (and of which I am the senior author)! The authors say they have addressed this issue by adding all relevant references - but they do so in a very elliptical manner and only in the Discussion (page 13). There is certainly

no explicit reference to the zebrafish studies, which constitute the most comprehensive in vivo analysis of the role of GRK2 in Hh signalling to date. Of course, the authors are careful to refer to "mammalian" Hh signalling throughout their report - but they should remember that Hh signalling was first elucidated in *Drosophila*, that Shh was first discovered in zebrafish and that chickens - which they use to validate the effect of GRK2 inhibition in vivo - are not mammals!!

Answer: We apologize and suggest that we would never overlook the seminal discoveries of others. We included a paragraph into Introduction of the revised manuscript, summarizing the known role of GRK2 in Hh signaling and placing the study in the context of published research (page 4, paragraph 3; page 5, paragraphs 1, 2). The study of Zhao et al (EMBO Reports, 2016) is cited in the revised manuscript's Introduction. The revised manuscript now cites 15 articles on the GRK2 role in Hh signaling, which are listed below. The original paragraph in Discussion summarizing the role of GRK2 in Hh was shortened in the revised manuscript (page 14, paragraph 3; page 15, paragraph 1), to avoid redundancy. The "mammalian" Hh signaling was changed to "vertebrate" Hh signaling throughout the revised manuscript.

References

Chen W, Ren X-R, Nelson CD, Barak LS, Chen JK, Beachy PA, de Sauvage F & Lefkowitz RJ (2004) Activity-Dependent Internalization of Smoothed Mediated by -Arrestin 2 and GRK2. *Science* (80-.). **306**: 2257–2260

Chen Y, Li S, Tong C, Zhao Y, Wang B, Liu Y, Jia J & Jiang J (2010) G protein-coupled receptor kinase 2 promotes high-level Hedgehog signaling by regulating the active state of Smo through kinase-dependent and kinase-independent mechanisms in *Drosophila*. *Genes Dev.* **24**: 2054–2067

Chen Y, Sasai N, Ma G, Yue T, Jia J, Briscoe J & Jiang J (2011) Sonic Hedgehog Dependent Phosphorylation by CK1 α and GRK2 Is Required for Ciliary Accumulation and Activation of Smoothed. *PLoS Biol.* **9**: e1001083

Desai PB, Stuck MW, Lv B & Pazour GJ (2020) Ubiquitin links smoothed to intraflagellar transport to regulate Hedgehog signaling. *J. Cell Biol.* **219**:

Evron T, Daigle TL & Caron MG (2012) GRK2: multiple roles beyond G protein-coupled receptor desensitization. *Trends Pharmacol. Sci.* **33**: 154–64

Evron T, Philipp M, Lu J, Meloni AR, Burkhalter M, Chen W & Caron MG (2011) Growth arrest specific 8 (Gas8) and G protein-coupled receptor kinase 2 (GRK2) cooperate in the control of smoothed signaling. *J. Biol. Chem.* **286**: 27676–27686

Gurevich E V, Tesmer JJG, Mushegian A & Gurevich V V (2012) G protein-coupled receptor kinases: more than just kinases and not only for GPCRs. *Pharmacol. Ther.* **133**: 40–69

Happ JT, Hedeem DS, Zhu J-F, Capener JL, Klatt Shaw D, Deshpande I, Liang J, Xu J, Stubben SL, Walker MF, Krogan NJ, Grunwald DJ, Hüttenhain R & Myers BR (2020) Smoothed Transduces Hedgehog Signals via Activity-Dependent Sequestration of 1 PKA Catalytic Subunits 2 3 Corvin D. *bioRxiv*: 2020.07.01.183079

Li S, Li S, Han Y, Tong C, Wang B, Chen Y & Jiang J (2016) Regulation of Smoothed Phosphorylation and High-Level Hedgehog Signaling Activity by a Plasma Membrane Associated Kinase. *PLoS Biol.* **14**:

Maier D, Cheng S, Faubert D & Hipfner DR (2014) A Broadly Conserved G-Protein-Coupled Receptor Kinase Phosphorylation Mechanism Controls *Drosophila* Smoothed Activity. *PLoS Genet.* **10**:

Meloni AR, Fralish GB, Kelly P, Salahpour A, Chen JK, Wechsler-Reya RJ, Lefkowitz RJ & Caron MG (2006) Smoothed signal transduction is promoted by G protein-coupled receptor kinase 2. *Mol. Cell. Biol.* **26**: 7550–60

Pal K, Hwang S hee, Somatilaka B, Badgandi H, Jackson PK, DeFea K & Mukhopadhyay S (2016) Smoothed determines β -arrestin-mediated removal of the G protein-coupled receptor Gpr161 from the primary cilium. *J. Cell Biol.* **212**: 861–875

Philipp M, Fralish GB, Meloni AR, Chen W, MacInnes AW, Barak LS & Caron MG (2008) Smoothed signaling in vertebrates is facilitated by a G protein-coupled receptor kinase. *Mol. Biol. Cell* **19**: 5478–89

Pusapati G V., Kong JH, Patel BB, Gouti M, Sagner A, Sircar R, Luchetti G, Ingham PW, Briscoe J & Rohatgi R (2018) G protein-coupled receptors control the sensitivity of cells to the morphogen Sonic Hedgehog. *Sci. Signal.* **11**: eaao5749

Zhao Z, Lee RTH, Pusapati G V, Iyu A, Rohatgi R & Ingham PW (2016) An essential role for Grk2 in Hedgehog signalling downstream of Smoothed. *EMBO Rep.* **17**: 739–752

Comment: My second reservation concerns the authors persistent failure to address the paradoxical finding that, despite the apparent loss of SMO activity in GRK2 mutant cells, the patient from whom these cells are derived developed to term without manifesting any of the gross defects characteristic of loss of SMO (or SHH). While the authors acknowledge that "This is an interesting comment" and affirm that "there were no structural CNS abnormalities noted in the three affected individuals", they fail to confront this issue in their discussion. As I pointed out the authors do hint at the possibility of a specific role for GRK2 in skeletogenesis, but this is not explored further. By this I meant that they did not consider why loss of GRK2 might specifically affect chondrocytes whilst sparing neural progenitors (and likely other cells). One fact that the authors overlook is that, unlike the zebrafish, mammals possess two apparently functionally redundant GRK2 paralogues, which explains why the mouse grk2 null phenotype is less extreme than that of the zebrafish. The authors might wish to consider the possibility that their GRK2 mutant chondrocyte lines are compromised in Hh signalling because they do not express the paralogous (GRK3) gene.

Answer: We thank the referee for pointing this out. The published studies show that deletion of Grk2 in the mouse leads only to mild defects in the neural tube patterning (Philipp et al., 2008), and that loss of both Grk2 and Grk3 (or their inhibition by the Grk2/3 inhibitor CMPD101) is required to fully inhibit the GLI3 processing and subsequently the Hh target gene expression in the murine neural precursor cells (NPCs) (Pusapati et al., 2018). This suggests at least partially redundant functions of Grk2 and Grk3 in cells and tissues expressing both proteins, and could explain no gross CNS abnormalities observed in the GRK2-null patients. Unlike NPCs, the mouse embryonic fibroblasts and NIH3T3 cells do not express Grk3 (Pusapati et al., 2018; Zhao et al., 2016); a genetic ablation of Grk2 alone is sufficient to obtain full inhibition of the Hh pathway in these cells. In line with these observations, the complete inhibition of the SAG-mediated GLI3 processing and *GLI1* and *PTCH1* expression observed in the GRK2-null patient skin fibroblasts suggest they either do not express GRK3 or that GRK3 does not contribute to Hh signaling in these cells. The reviewer brings up an excellent point on the topic. We have previously published work on cartilage gene expression determined by RNA seq (Li et al., 2017) compared to a total of twenty human tissues. GRK2 is ubiquitously expressed throughout these tissues, with robust levels of expression in both chondrocytes and brain. However, GRK3 has very, very, low level of expression in chondrocytes with higher levels in brain by 4 fold. As suggested, the paralog GRK3 may have a more substantial role in neurodevelopment, but not in cartilage, supporting the skeletal phenotype due to loss of GRK2. It is of note however, that Grk3

knockout mice develop normally (Peppel et al, 1997). The revised manuscript now contains a paragraph discussing the Grk3 (page 15, paragraph 2).

References:

Li B, Balasubramanian K, Krakow D, Cohn DH. (2017) Genes uniquely expressed in human growth plate chondrocytes uncover a distinct regulatory network. *BMC Genomics*. **18**:983

Peppel K, Boekhoff I, McDonald P, Breer H, Caron MG & Lefkowitz RJ (1997) G protein-coupled receptor kinase 3 (GRK3) gene disruption leads to loss of odorant receptor desensitization. *J. Biol. Chem.* **272**: 25425–25428

Philipp M, Fralish GB, Meloni AR, Chen W, MacInnes AW, Barak LS & Caron MG (2008) Smoothed signaling in vertebrates is facilitated by a G protein-coupled receptor kinase. *Mol. Biol. Cell* **19**: 5478–89

Pusapati G V., Kong JH, Patel BB, Gouti M, Sagner A, Sircar R, Luchetti G, Ingham PW, Briscoe J & Rohatgi R (2018) G protein-coupled receptors control the sensitivity of cells to the morphogen Sonic Hedgehog. *Sci. Signal.* **11**: eaao5749

Zhao Z, Lee RTH, Pusapati G V, Iyu A, Rohatgi R & Ingham PW (2016) An essential role for Grk2 in Hedgehog signalling downstream of Smoothed. *EMBO Rep.* **17**: 739–752

Referee #2 (Comments on Novelty/Model System for Author)

The existing animal models are early embryonic lethal and do not allow downstream analyses. The authors are working to develop conditional mouse model but this will be beyond the scope of this study. Given these limitations, the authors provide abundant in vitro data that is well-controlled and recapitulated in several different cell culture models derived from mouse or human samples, using different approaches including knock-down experiments and chemical inhibition, and supported by in vivo/ex vivo chick limb bud model. This reviewer finds the above model systems and experimental approaches to be adequate and sufficient to support the authors hypothesis.

Referee #2 (Remarks for Author)

Summary: The authors submitted a revised manuscript reporting pathogenic variants in GRK2 as the cause of new form of asphyxiating thoracic dystrophy (ATD), a genetically and clinically heterogeneous disorder. In this revised version the authors provided abundant experimental evidence to support the role of GRK2 in skeletogenesis, and to demonstrate that loss of GRK2 is associated with ciliary dysfunction and dysregulated Hh and Wnt signaling pathways contributing to the patients' phenotype. The manuscript is well-written, and contains novel data that is clinically relevant and of broad significance. The authors appropriately addressed this reviewer's concerns, and I only have a few minor suggestions that may make the manuscript better.

Minor comments

Comment 1: The authors should consider revising Figure 3 (such that the "flow" of data is consistent with the A to F organization). The abundant data is appreciated but difficult to follow in the current format.

Answer: We thank the referee for pointing this out. The flow of the Figure 3 was changed to follow the A to F organization.

Comment 2: In Figure 5B and EV2B the quantification of SMO signal still suggests some translocation of SMO to the cilia even in the presence of inhibitors (that is statistically

significant when compared to cells without SAG treatment). I suggest to change the title to express that inhibition of GRK2 activity partially inhibited, or decreased, rather than completely blocked Smoothed accumulation in the cilia. This will also be consistent with Hh impaired signaling not being the only player but rather one of the causes of abnormal skeletal development, along with dysregulated Wnt signaling (and possibly other pathways as mentioned in the discussion).

Answer: In agreement with the referee's comment, the figure legend titles for Figure 5 and EV2 were rephrased to reflect the fact that inhibition of GRK2 activity decreases the ciliary accumulation of SMO.

Comment 3: In Figure EV1A please clarify in the bottom panel whether red staining marks pericentrin or acetylated tubulin.

Answer: In the bottom panel of the Figure EV1A, the red staining marks both acetylated tubulin and pericentrin. This combination was used specifically for this panel to clearly indicate the cilia base in order to properly localize GLI3 to the ciliary tips. To calculate the percentage of ciliated cells in panel (C), no pericentrin counterstaining was used, and so no structure labeled by a red fluorochrome in addition to the axonemal acetylated tubulin could have interfered with the analysis.

Comment 4: In Figure EV4 while panels C, D and E appear to be similarly treated the graphic presentation is not consistent across different models (specifically the +/- paroxetine and Wnt3A treatments are organized differently in the RCS panel as compared to human chondrocyte cells). This is somewhat confusing and I suggest to revise panel C to match with panels D, E below.

Answer: In agreement with the referee, we reorganized the blots and graphs in Figure EV4C to match the flow of the panels D, E.

Editor comments

Comment: In the main manuscript file, please do the following:

- Data availability: we notice that you have submitted the patient data to dbGAP. Please provide accession number in this section.

Answer: Thank you for your helpful comment and we apologize for any confusion regarding the submission of the variant p.Arg158* to dbGAP. This patient was consented in 2005 under a human subject consent that predated the creation of dbGAP and thus we did not include it our submissions to dbGAP. Thus, none of these variants have been submitted to dbGAP. We appreciate your understanding.

We are pleased to inform you that your manuscript is accepted for publication and is now being sent to our publisher to be included in the next available issue of EMBO Molecular Medicine.

Corresponding Author Name: Pavel Krejci, Deborah Krakow

Journal Submitted to: EMBO Mol Med

Manuscript Number: EMM-2019-11739